# Functional control of heteromeric Kv2.1/6.4 channels by the voltage sensor domain of the silent Kv6.4 subunit

Debanjan Tewari [ID], Christian Sattler [ID] and Klaus Benndorf [ID]

*Institut für Physiologie II, Universitätsklinikum Jena, Friedrich-Schiller-Universität Jena, Jena, Germany*

Handling Editors: Peying Fong & Brian Delisle

The peer review history is available in the Supporting Information section of this article (https://doi.org/10.1113/JP288376#support-information-section).

**Abstract figure legend** Schematic illustration of the gating transitions of the Kv2.1–Kv6.4 heteromeric potassium channel. The figure depicts three functional states: the closed state (top left), in which the channel is non-conducting (the S2 segment (green helix) and a phenylalanine residue (Phe, blue circle) in Kv6.4 are indicated); the open state (top right): upon activation, the channel undergoes a conformational change, allowing ion conduction; and the inactivated state (bottom right): the channel can transition either from the open or the closed state to the inactivated state. The transition from the closed state is shown (blue arrow). Bottom left: illustration of a representative current trace.

**Abstract** In the activation process of Kv channels, the S4 segment of the voltage-sensing domain (VSD) moves in the outward direction. A conserved phenylalanine in the transmembrane S2 helix of the VSD is viewed as operating as a charge transfer centre (CTC) that interacts with a positively charged arginine of the S4 helix. This phenylalanine is highly sensitive to diverse substitutions. Kv2.1 subunits can form functional homotetrameric channels on their own whereas 'silent' Kv6.4 subunits can only contribute to functional heterotetrameric channels. We used concatenated dimers of Kv2.1 and Kv6.4 subunits to define the stoichiometry and position of these subunits in functional heterotetrameric channels. Our results demonstrate that mutating the phenylalanine F273 of the Kv6.4 subunits in Kv 2.1_6.4 channels built of dimers to diverse other amino acids at the CTC affects steady-state activation only moderately whereas it strongly shifts steady-state inactivation by 40 mV toward more depolarized potentials compared to Kv2.1_6.4 wild-type channels. Mutating the Kv6.4 subunits in this heterotetramer slowed down the recovery from closed-state inactivation without impacting open-state inactivation. Moreover, results with the specific Kv2.1 blocker guangxitoxin suggest that Kv6.4 subunits may partly activate Kv2.1_6.4 channels. It is concluded that F273 in the silent Kv6.4 subunit of Kv2.1_6.4 channels has a unique role in controlling activation and the recovery from inactivation.

(Received 19 December 2024; accepted after revision 12 May 2025; first published online 4 June 2025)

**Corresponding authors** Debanjan Tewari and Klaus Benndorf, Institut für Physiologie II, Universitätsklinikum Jena, Friedrich-Schiller-Universität Jena, 07740 Jena, Germany. Email: Debanjan.Tewari@med.uni-jena.de and Klaus.Benndorf@med.uni-jena.de

## Highlights

- This study quantifies the functional effects of Kv6.4 mutations in Kv2.1_6.4 channels on activation and inactivation.
- Highly diverse mutations of the phenylalanine in the charge transfer centre of Kv6.4 reveal its unique role in Kv2.1_6.4 channels in closed state inactivation.
- The specific Kv2.1 blocker guangxitoxin unmasks that Kv6.4 subunits can partly activate Kv2.1_6.4 channels.

## Introduction

The human genome encodes what are presumed to be 78 genes for potassium ion channels. Among them, the voltage-gated potassium (Kv) channel superfamily is largest, comprising about 40 genes (Gutman et al., 2005; Maljevic & Lerche, 2013; Wulff et al., 2009). In concert with other channels, Kv channels control the excitability of various cell types, such as neurons and cardiac myocytes (Wulff et al., 2009), by maintaining the resting membrane potential, enabling repolarization, shaping action potentials and regulating neuronal firing rates (Blaine & Ribera, 2001; Guan et al., 2013; Johnston et al., 2010).

Kv channels fulfil their biological functions by transitioning between closed, open and inactivated states in response to membrane voltage (Bähring & Covarrubias, 2011; Hoshi et al., 1990). Structurally, these channels can exist as either homotetramers, composed of identical subunits, or heterotetramers, formed by homologous subunits. In homotetramers, four subunits assemble into the channel. Each subunit contains six trans-

**Debanjan Tewari** earned his PhD from the Indian Institute of Technology, Madras, India, where he developed a strong interest in electrophysiology and ion channel research. He is currently working as a research associate in the laboratory of Prof. Dr Klaus Benndorf at the Institute of Physiology II, University Hospital Jena. Here, his research primarily focuses on voltage-gated ion channels, with a particular emphasis on the potassium (Kv) channel family. In addition to his work on voltage-gated potassium channels, he is also interested in HCN ion channels.

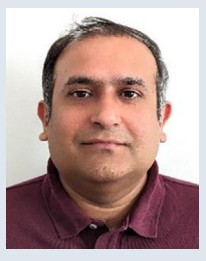

membrane segments. The first four segments (S1–S4) make up the voltage-sensing domains (VSDs), which detect changes in membrane potential. The S4 segment, enriched with positively charged residues, serves as the primary voltage sensor sequence within the VSD. The remaining segments, S5 and S6, align around a central axis to form the ion-conducting pore (Doyle et al., 1998; Long et al., 2005a, 2005b). When the membrane depolarizes, the S4 segments undergo a combination of rotational, tilting and vertical movements, producing gating currents observable in various Kv and other voltage-gated channels (Bezanilla, 2008; Bezanilla et al., 1991). These conformational changes are transmitted through electromechanical coupling to an intracellular channel gate, evoking channel opening. This gate is formed by the C-terminal ends of the four S6 transmembrane segments, which, in the closed state, block the central ion-conducting pore through a bundle-crossing configuration (Long et al., 2005a; Lu et al., 2002).

It has been shown that the F233 residue in the chimeric Kv1.2/2.1 channel functions as a hydrophobic plug. This residue is highly conserved across the voltage-sensing domains (VSDs) of various voltage-gated channels, including Kv, Nav and Cav channels, suggesting a major role in channel gating (Long et al., 2007). Its importance for activation was first demonstrated by mutational analysis and conductance–voltage ($G$–$V$) plots (Li-Smerin et al., 2000). More recently, it was demonstrated that a rigid, cyclic side chain at the F233 position is crucial for gating by interacting sequentially with all four gating charges (Tao et al., 2010). Later, gating current measurements of various F233 mutants revealed that a hydrophobic residue at this position primarily regulates the transfer of the final gating charge across F233, while the movement of initial charges remains unaffected (Lacroix & Bezanilla, 2011).

Within the Kv superfamily, Kv1-4 subunits can form functional homotetramers. In contrast, 10 so-called silent subunits (KvS) cannot independently assemble into functional channels. In mammals, these silent subunits are Kv5.1, Kv6.1–6.4, Kv8.1–8.2 and Kv9.1–9.3 (Bocksteins, 2018; Bocksteins & Snyders, 2012). Notably, also the KvS subunits contain the conserved phenylalanine at the charge transfer centre. While unable to form functional channels on their own, KvS subunits can integrate into heteromeric channels, imparting unique biophysical properties as compared to homotetrameric channels formed by Kv2 subunits (Bocksteins, 2018). For example, co-expression of Kv6.4 with either Kv2.1 or Kv2.2 revealed that heteromers of Kv2.1 and Kv6.4 exhibit a 40 mV shift in steady-state toward more negative potentials and a nearly fivefold reduction in current density compared to Kv2.1 homomers. Additionally, voltage-dependent activation in these heteromers is less steep than in Kv2.1 homomers, and the activation time

course is more complex (Bocksteins, 2016). Recently, Stewart et al. also demonstrated that guangxitoxin-1E selectively inhibits the Kv2.1/6.4 current, providing a method to pharmacologically separate the conductance of Kv2.1/KvS channels (Stewart et al., 2024).

In many Kv channels, sustained depolarization triggers a slow inactivation process involving changes in the selectivity filter, resulting in a non-conductive state. Interestingly, another inactivation can sometimes occur even before the intracellular channel gate opens, known as closed-state inactivation (CSI) (Bähring & Covarrubias, 2011; Bähring et al., 2012). The silent Kv6.4 subunit has been shown to induce CSI in heteromeric Kv2.1–Kv6.4 channels. However, the molecular mechanism underlying the induction of closed-state inactivation in heteromeric Kv2.1–Kv6.4 channels is still elusive, which is likely due to only vague knowledge about the orientation and stoichiometry in the heteromers if compared to other channel families.

Herein, we aim to further clarify the role of the Kv6.4 subunit in governing the activation and inactivation of the heteromeric Kv2.1–Kv6.4 channel. We use various dimer constructs of wild-type (WT) and mutated subunits and investigated at defined subunit stoichiometry the effect on channel gating of F273 in the charge transfer centre of the silent Kv6.4 subunit. We show that this amino acid 273 in the silent Kv6.4 subunit is unique in controlling activation and the recovery from inactivation of heterotetrameric Kv2.1_6.4 channels.

## Methods

### Ethical approval

Oocytes were obtained from Ecocyte (Castrop-Rauxel, Germany) or surgically harvested from anaesthetized *Xenopus laevis* frogs using 0.1% tricaine (pH 7.1; MS-222, Parmaq, Fordingbridge, UK). All procedures involving *Xenopus laevis* were approved by the Friedrich Schiller University Jena Animal Ethics Committee (UKJ-18-008, 9 May 2018 and UKJ-23-005, 28 April 2023). Frog housing conditions were maintained at 17–21°C room temperature, 18–22°C water temperature, and 52% humidity, with a 12-h light cycle. Environmental enrichment included black shelters, and frogs were fed twice weekly (~2 teaspoons per tank). Oocytes were extracted under tricaine anaesthesia (pH 7.0–7.5) for 5–10 min, with anaesthesia depth confirmed via the toe-pinch reflex. Frogs deemed unsuitable for further surgeries were euthanized by blunt head trauma followed by decapitation. Procured oocytes were incubated in $Ca^{2+}$-free solution (82.5 mM NaCl, 2 mM KCl, 1 mM $MgCl_2$, 5 mM Hepes, pH 7.4) containing 3 mg/ml collagenase A (Roche, Grenzach-Wyhlen, Germany) for 105 min. Stage IV and V oocytes were manually

defolliculated, injected with 30 nl mRNA (0.2 μg/μl), and incubated at 18°C in Barth's medium for 24 h to allow ion channel synthesis and trafficking to the plasma membrane.

## Molecular biology

The genes encoding human Kv2.1 (*KCNB1*) and Kv6.4 (*KCNG4*) were inserted upstream of a T7 promoter in the pGEMHE vector, as described previously (Gusic et al., 2021). Using recombinant DNA cloning techniques, a concatenated Kv2.1–Kv6.4 dimer was generated. The Kv2.1 subunit was amplified with primers incorporating a *Kpn*I restriction site before the coding region and an *Asc*I site after an introduced linker sequence. Similarly, the Kv6.4 subunit was amplified using primers with an *Asc*I site within the linker sequence and an *Xba*I site following the stop codon. The start codon in Kv6.4 and the stop codon in Kv2.1 were removed to ensure continuity. A 10-amino-acid linker sequence (KARPTEGSLA) was introduced between the two subunits. For controls a Kv2.1–Kv2.1 dimer with the same linker and a Kv6.4–Kv2.1 dimer with an alternative linker (GTRIPGKQLS) were constructed. Mutations were introduced via overlapping PCR and subcloning of

specific fragments. The resulting clones were validated through restriction analysis and sequencing (Microsynth SEQLAB, Göttingen, Germany). cRNA for injection was synthesized using the mMACHINE T7 kit (Thermo Fisher Scientific, Waltham, MA, USA), and its quality was confirmed by gel electrophoresis.

## Electrophysiology, data analysis and statistics

Whole-cell currents from Kv2.1, Kv6.4 and concatemeric constructs were recorded using a two-electrode voltage clamp (725C amplifier, Warner Instruments, LLC, Hampden, CT, USA) at room temperature. Glass electrodes (Brand GmbH + Co KG, Wertheim, Germany) were filled with 3 M KCl solution, with resistances ranging from 0.3 to 0.9 MΩ.

The bath solution consisted of 96 mM NaCl, 2 mM KCl, 1.8 mM CaCl$_2$, 1 mM MgCl$_2$, and 10 mM Hepes (pH 7.4). Activation of currents was elicited by test pulses ranging from −100 to +100 mV in 10 mV increments from a holding potential of −90 mV, with each test pulse lasting 300 ms inactivation was studied with pre-pulse potentials from −120 mV to +60 mV in 10 mV increments from a holding potential of −90 mV, with a 5-s test pulse duration and an inter-pulse interval

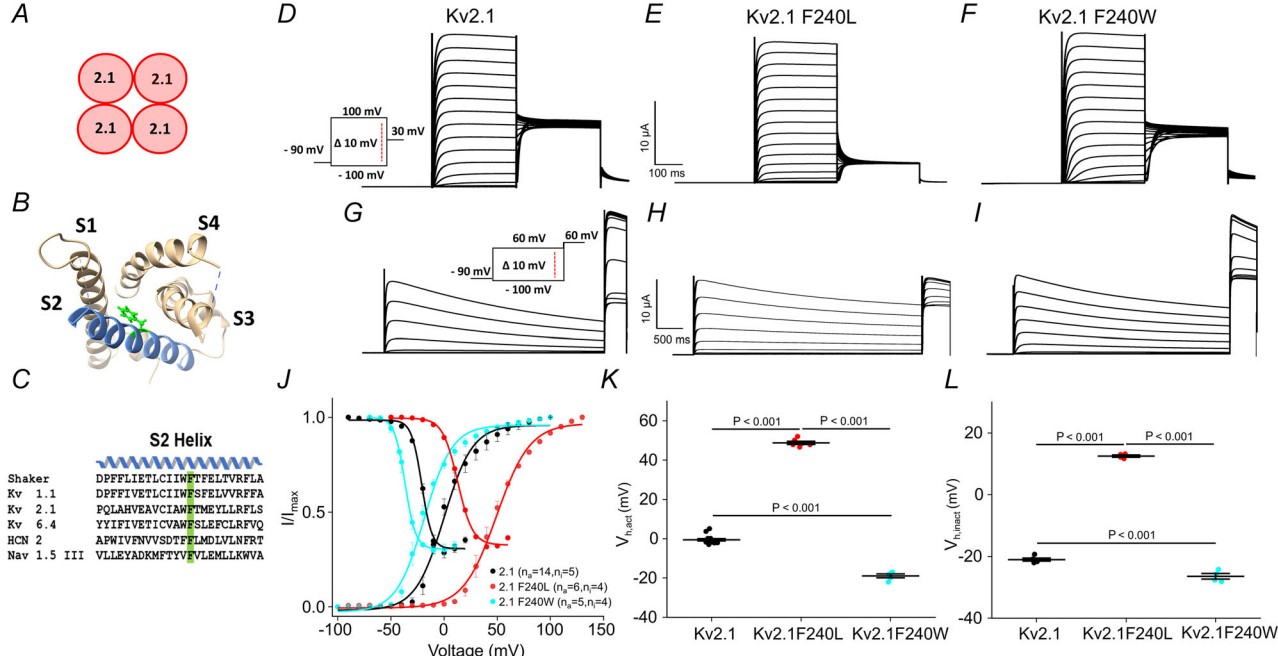

**Figure 1. Voltage-dependent activation and inactivation of Kv2.1 and its CTC mutants**
*A*, cartoon of a homomeric Kv2.1 channel. *B*, structure of a Kv2.1 VSD from S1 to S4 (PDB ID 8SD3). The S2 helix with the conserved phenylalanine is highlighted in green. *C*, sequence alignment of the S2 helix with highlighted charge transfer centre in green. *D–I*, representative current traces for activation and inactivation of the WT Kv2.1, F240L and F240W mutant channels. *J*, steady-state activation and inactivation relationships for homomeric Kv 2.1 and mutant channels were obtained from tail currents. *K*, boxplot $V_{h,act}$. *L*, boxplot $V_{h,inact}$. Data points represent means ± SD, fitted to a Boltzmann equation. [Colour figure can be viewed at wileyonlinelibrary.com]

**Table 1. Fit parameters of the constructs with a Boltzmann equation**

| Construct | $n_a$ | $V_{h,act}$ (mV) | $k_{act}$ (mV) | $n_i$ | $V_{h,inact}$ (mV) | $k_{inact}$ (mV) |
|---|---|---|---|---|---|---|
| Monomers | | | | | | |
| 2.1 | 14 | $-0.5 \pm 2$ | $12.9 \pm 1$ | 5 | $-21.0 \pm 0.5$ | $4.8 \pm 0.2$ |
| 2.1 F240L | 6 | $48.8 \pm 2$ | $14.7 \pm 1$ | 4 | $12.7 \pm 0.3$ | $5.0 \pm 0.4$ |
| 2.1 F240W | 5 | $-18.9 \pm 2.1$ | $13.2 \pm 0.5$ | 4 | $-26.4 \pm 1.2$ | $5.0 \pm 0.5$ |
| 2.1/6.4 | 8 | $-12.9 \pm 3.5$ | $18.0 \pm 1.6$ | 4 | $-58.8 \pm 0.5$ | $9.9 \pm 0.9$ |
| 2.1/6.4L | 5 | $-3.4 \pm 1$ | $12.6 \pm 1.6$ | 4 | $-11.9 \pm 1.3$ | $5.3 \pm 0.6$ |
| 2.1/6.4W | 4 | $-2.2 \pm 3.5$ | $15 \pm 1.7$ | 4 | $-8.6 \pm 0.5$ | $5.3 \pm 0.9$ |
| Dimer | | | | | | |
| 2.1_2.1 | 5 | $6.4 \pm 2$ | $14.5 \pm 0.4$ | 3 | $-18.1 \pm 0.5$ | $6.5 \pm 0.5$ |
| 2.1_6.4 | 5 | $-13.9 \pm 2$ | $16.7 \pm 0.5$ | 7 | $-64 \pm 2.3$ | $6 \pm 0.6$ |
| 6.4_2.1 | 6 | $-13.7 \pm 2.3$ | $15.8 \pm 0.3$ | 5 | $-62.9 \pm 1.4$ | $7 \pm 0.8$ |
| 2.1_6.4A | 5 | $0.80 \pm 2.7$ | $13.0 \pm 1.4$ | 4 | $-20 \pm 1$ | $5.8 \pm 0.3$ |
| 2.1_6.4V | 4 | $-2.7 \pm 1$ | $13.3 \pm 1$ | 5 | $-22.5 \pm 1.6$ | $5 \pm 0.6$ |
| 2.1_6.4L | 6 | $-3.6 \pm 0.3$ | $12.0 \pm 0.3$ | 4 | $-20 \pm 1.5$ | $5.4 \pm 0.5$ |
| 2.1_6.4IL | 6 | $3.8 \pm 2.5$ | $11 \pm 0.7$ | 6 | $-16.4 \pm 1.5$ | $6.3 \pm 0.8$ |
| 2.1_6.4D | 4 | $-2.8 \pm 1.2$ | $13.5 \pm 2$ | 4 | $-26.8 \pm 3.5$ | $5.2 \pm 0.5$ |
| 2.1_6.4E | 4 | $14 \pm 1.7$ | $13.6 \pm 2$ | 4 | $-15.1 \pm 1.2$ | $5.7 \pm 0.2$ |
| 2.1_6.4K | 9 | $-2.6 \pm 1.3$ | $9.5 \pm 4.2$ | 5 | $-21.9 \pm 1.5$ | $5.0 \pm 0.6$ |
| 2.1_6.4R | 5 | $2 \pm 2.5$ | $14.7 \pm 0.7$ | 4 | $-15.2 \pm 1.3$ | $5.7 \pm 0.3$ |
| 2.1_6.4Y | 4 | $-10 \pm 2.4$ | $13 \pm 1.5$ | 4 | $-25.1 \pm 1.1$ | $5.5 \pm 0.2$ |
| 2.1_6.4W | 5 | $2 \pm 1$ | $14.7 \pm 1$ | 4 | $-04.6 \pm 1.5$ | $6.1 \pm 0.9$ |

Values are given as means $\pm$ SD. The number of oocytes analysed for each construct is given in the table. The midpoints of activation and inactivation ($V_{h,act}$ and $V_{h,inact}$) and the slope factors ($k_{act}$ and $k_{inact}$) were obtained by fitting Equation (1).

of 10–15 s. All currents were sampled at 5.00 kHz. Data acquisition and preliminary analysis were performed using Patchmaster, Fitmaster software (HEKA Elektronik GmbH, Reutlingen, Germany) and finally on OriginPro 2019 software (OriginLab Corp., Northampton, MA, USA). Steady-state activation was determined from the amplitude of instantaneous tail currents at +30 mV. The experimental data points were fitted with a single equation according to:

$$I/I_{max} = 1/(1 + \exp((V - V_{h,act})/k_{act})) \qquad (1)$$

where $V_{h,act}$ is the voltage of half-maximum activation, $k_{act}$ the slope factor, $V$ the membrane potential, and $I_{max}$ the maximum current. For fitting steady-state inactivation, Eqn (1) was used accordingly, yielding $V_{h,inact}$ and $k_{inact}$. In the presence of guangxitoxin-1E, steady-state activation was fitted with sum of two Boltzmann functions according to

$$I/I_{max} = A/(1 + \exp(zd_1F(V - V_{h1})/RT)) + (1 - A)/$$
$$(1 + \exp(zd_2F(V - V_{h2})/RT)) \qquad (2)$$

Here, $V_{h1}$ and $V_{h2}$ as well as $zd_1$ and $zd_2$ have the corresponding meaning for the two components. $A$ is a calibration factor. F is the Faraday constant.

The time constants of activation were determined by fitting the activation time course by a single exponential function. The recovery from inactivation was determined

by fitting the exponential function $y = y_0 + A_1 \times (1 - \exp(-t/\tau_{rec}))$ to the recovery time courses using OriginPro 2019, where $y$ is the normalized amplitude of the test current, $y_0$ a time-independent component, $t$ the time and $\tau_{rec}$ the recovery time constant.

Data represent means $\pm$ SD. For all channel parameters, such as half-activation ($V_{h,act}$) and half-inactivation voltages ($V_{h,inact}$) and recovery time constants ($\tau_{rec}$), we used one-way ANOVA analysis. $n$ and $P$ values are mentioned in the text as well as figures.

## Results

### Impact of F240 voltage-dependent activation and inactivation in Kv2.1

Based on previous reports (Schwaiger et al., 2013; Tao et al., 2010), the charge transfer centre of wild-type Kv 2.1 (Kv 2.1 WT) was subjected to mutagenesis, replacing the highly conserved phenylalanine at position 240 (F240) of the S2 helix with leucine and tryptophan as shown in the structure using chimera as shown in Fig. 1A–C (Fernández-Mariño et al., 2023; Pettersen et al., 2021). Activation and inactivation of the currents were measured as shown by the protocols in Fig. 1D–I. Analysis of the $I/I_{max}$ *versus* voltage relationship shows the half-activation voltage ($V_{h,act}$) for Kv2.1 WT ($n_a = 14$) is $-0.5 \pm 2$ mV,

consistent with a previous report (Tewari et al., 2024). Substitution of F240 with leucine (F240L, $n_a = 6$) resulted in a rightward shift of the $I/I_{max}$ curve by more than 40 mV, indicating that this mutation increases the voltage required for channel activation as shown in Fig. 1*J* and *K*. Conversely, replacement with tryptophan (F240W, $n_a = 5$) induced a leftward shift of activation, suggesting a reduction in the voltage threshold for activation. These shifts were quantified by fitting the activation curves ($I/I_{max}$) to a Boltzmann equation (Table 1). The observed alterations in $V_{h,act}$ highlight the critical role of the F240 residue in modulating the gating properties of Kv2.1. Further, the inactivation was measured for all three constructs (Fig. 1*G–I*). To analyse the inactivation properties, the normalized tail currents ($I/I_{max}$) were plotted as a function of voltage, and the data were fitted to a Boltzmann equation to determine the half-inactivation voltage ($V_{h,inact}$) mentioned in Fig. 1*J*. For the wild-type Kv2.1 channel ($n_i = 5$), the $V_{h,inact}$ was determined to be $-21 \pm 0.5$ mV. In comparison, the leucine mutant ($n_i = 4$) exhibited a rightward shift in the $V_{h,inact}$, with a value of $12.7 \pm 0.3$ mV. Conversely, the tryptophan mutant ($n_i = 4$) displayed a leftward shift with a value of $-26.4 \pm 1.2$ mV compared to the wild-type (Fig. 1*L*).

## Effects of charge transfer centre mutations in the Kv 6.4 subunit

To investigate the functional role of Kv6.4, an experiment was conducted wherein the mRNAs of Kv2.1 and Kv6.4 were mixed in a 1:1 ratio and co-expressed in *Xenopus laevis* oocytes (Lacroix et al., 2024). The resulting heteromeric Kv2.1–Kv6.4 channel complexes were studied using a voltage-clamp protocol as described in Fig. 2*A–F*. Electrophysiological recordings revealed significant alterations in the voltage-dependent properties of the heteromeric channels compared to the Kv2.1 homomeric channels. Notably, both $V_{h,act}$ and $V_{h,inact}$ were shifted in the Kv2.1–Kv6.4 heteromer relative to Kv 2.1 homomers (Fig. 2*G–I*). Of particular interest is a pronounced shift in inactivation for the Kv2.1–Kv6.4 heteromeric channel (Table 1), suggesting that the incorporation of Kv6.4 subunits substantially alters the inactivation gating of the heteromeric channel, a well-established result (Bocksteins, 2016, 2018; Bocksteins & Snyders, 2012; Möller et al., 2020).

To understand the above-mentioned effect in detail, the leucine, which shows a reduced sensitivity to voltage in the Kv 2.1 background, and tryptophan, which shows an

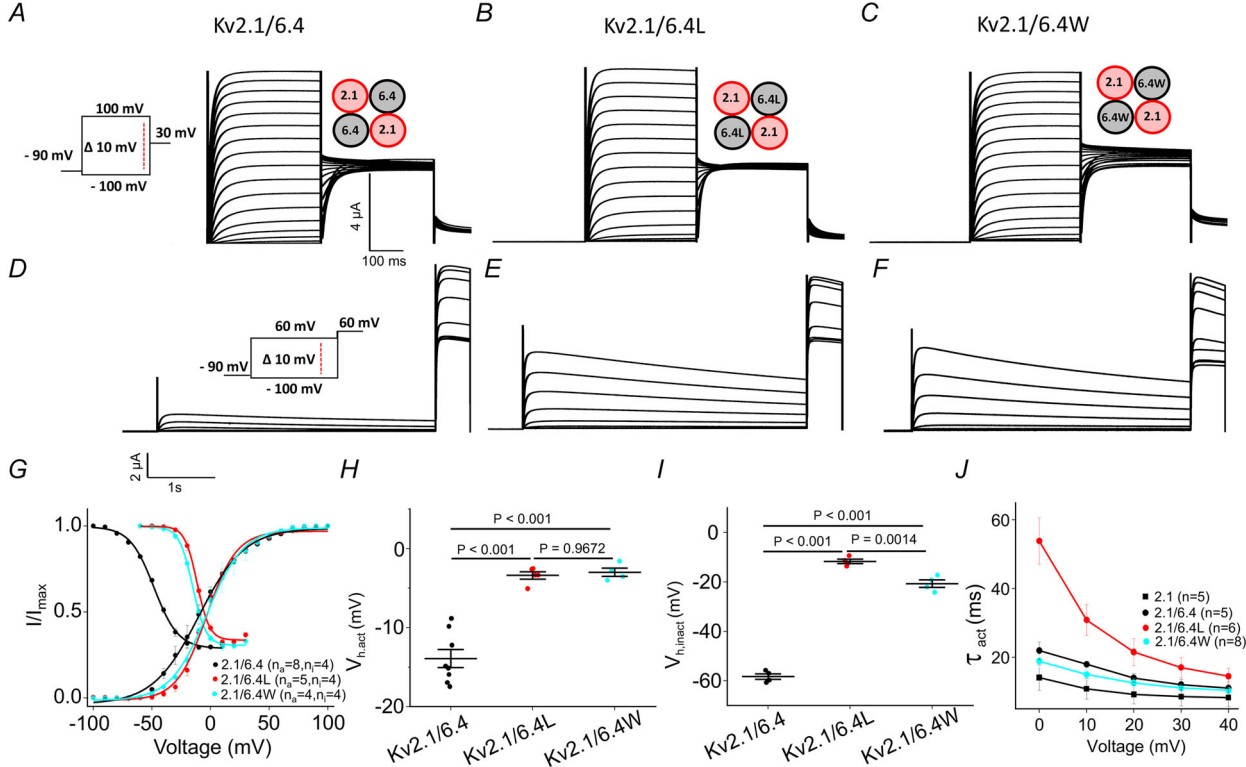

**Figure 2. Biophysical characterization of Kv2.1/6.4 monomer**
*A*, cartoon of Kv2.1/6.4 monomer. *B–D*, representative current traces for determining steady-state activation in the three indicated channels formed by co-expressed monomers. *E–G*, representative families of currents used to determine steady-state inactivation for these channels. *H*, corresponding steady-state activation and inactivation relationships. *I*, boxplot $V_{h,act}$. *J*, boxplot $V_{h,inact}$. *K*, activation time constant as function of voltage. [Colour figure can be viewed at wileyonlinelibrary.com]

increased sensitivity to voltage in the Kv2.1 background, were introduced at the charge transfer centre of the Kv6.4 subunit (F273). The effects of these mutations on channel properties were examined by co-expressing the mutated Kv6.4 subunits with Kv2.1. Significant changes were observed in both activation and inactivation compared to the Kv2.1/6.4 heteromeric channel. The $V_{h,act}$ of the activation curve for the Kv2.1/6.4 ($n_a$ = 8) channel was −12.9 ± 3.5 mV. However, the leucine ($n_a$ = 5) and tryptophan ($n_a$ = 4) mutant channels exhibited depolarized activation, with $V_{h,act}$ values of −3.4 ± 1 and −2.2 ± 3.5 mV, respectively (Fig. 2*G* and *H* and Table 1). Furthermore, while the inactivation of Kv 2.1/6.4 ($n_i$ = 4) channel displayed a marked leftward shift ($V_{h,inact}$ = −58.8 ± 0.5 mV), inactivation of the mutant channels was shifted towards the right ($V_{h,inact}$ for leucine ($n_i$ = 4) and tryptophan ($n_i$ = 4) −11.9 ± 1.3 and −8.6 ± 0.5 mV, respectively as shown in Fig. 2*G* and *I*). These results demonstrate that mutations at the charge transfer centre of the Kv6.4 subunit substantially modulate the

functional properties of Kv2.1/6.4 heteromeric channels. When Kv2.1/6.4 channels are expressed in a heterologous system, they can form subunit ratios of either 2:2 or 3:1 (Möller et al., 2020; Pisupati et al., 2018), but recent studies suggest the channels predominantly assemble in a 2:2 ratio, with the silent subunits positioned opposite to each other rather than as neighbours (Möller et al., 2020).

## Use of Kv2.1_6.4 dimers

To preserve both the stoichiometry and the specific orientation, two subunits were concatenated via a linker, as indicated by a '_' in the following. The electrophysiological properties of the Kv2.1_2.1, Kv2.1_6.4 and Kv6.4_2.1 dimers were first analysed to determine whether the linker influences channel function (Fig. 3*A–F*). The steady-state activation and inactivation relationships for these dimers closely resemble those of the monomeric channels (Fig. 3*G*). Also, the activation time constants are consistent with the monomeric form shown in Fig. 3*H*. These results indicate that dimerization does not significantly affect the electrophysiological properties of the channel. Further, the dimer exhibited improved expression compared to expression of Kv 6.4_2.1 dimer and Kv 2.1/6.4 monomer, shown in Fig. 4.

The two mutations were subsequently introduced into the charge transfer centre of the Kv6.4 subunit (F273) of the Kv2.1_6.4 dimer and activation and inactivation were studied (Fig. 5*A–J*). The Kv2.1_6.4 dimer exhibited $V_{h,act}$ and $V_{h,inact}$ values comparable to those of the monomeric channel (Kv 2.1/6.4), as summarized in Table 1. While the $V_{h,act}$ values for the other two mutants remained like their monomeric counterpart some differences in $V_{h,inact}$ were observed when comparing the monomers and the dimers.

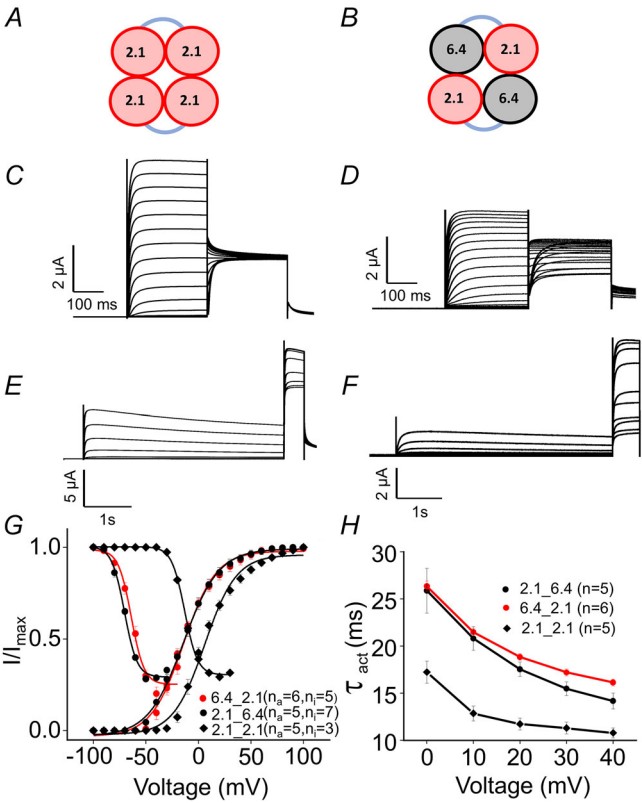

**Figure 3. Impact of linker on dimeric channel function**
*A* and *B*, cartoon of Kv2.1_2.1 and Kv6.4_2.1 dimer. *C* and *D*, example current traces showing activation of Kv2.1_2.1 and Kv6.4_2.1 dimers. *E* and *F*, representative traces illustrating inactivating current for the above-mentioned constructs. *G*, steady-state activation for Kv2.1, Kv2.1_6.4 and Kv6.4_2.1. *H*, time constant for activation ($\tau_{act}$) as function of voltage. Data points represent means ± SD for each voltage tested. [Colour figure can be viewed at wileyonlinelibrary.com]

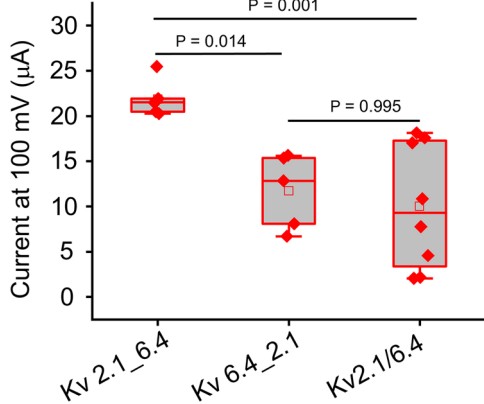

**Figure 4. Expression profile**
Quantification of current for Kv2.1_6.4, Kv 6.4_2.1 and Kv 2.1/6.4 constructs at 100 mV following the injection of 30 nl of mRNA at a concentration of 0.2 μg/μl into individual oocytes (approximately 6 ng), measured after 24 h of incubation. [Colour figure can be viewed at wileyonlinelibrary.com]

**Table 2. Effects of GTX on steady-state activation**

| Construct | $n_a$ | Without GTX-1E | GTX-1E | |
|---|---|---|---|---|
| | | $V_{h,act}$ (mV) | $V_{h,act}$ (mV) (1) | $V_{h,act}$ (mV) (2) |
| 2.1_2.1 | 3 | $0.05 \pm 0.6$ | $82.4 \pm 1.4$ | |
| 2.1_6.4 | 4 | $-13.9 \pm 0.9$ | $-7 \pm 1.4$ | $74.3 \pm 1.8$ |
| 2.1_6.4L | 4 | $-3.6 \pm 0.3$ | $0.2 \pm 0.7$ | $112.2 \pm 1.6$ |
| 2.1_6.4W | 5 | $2 \pm 1$ | $-5.5 \pm 1.8$ | $107.5 \pm 1.3$ |

Values are given as means $\pm$ SD. The number of oocytes analysed in the presence of the toxin is given in the table. The midpoints of activation ($V_{h,act}$) were obtained by fitting the $I/I_{max}$ relationship with the sum of two Boltzmann functions (Eqn (2)).

### Sensitivity of Kv2.1_Kv6.4 channels to guangxitoxin-1E

Earlier studies have demonstrated that Kv2.1 channels are sensitive to guangxitoxin-1E (Gupta et al., 2015; Tilley et al., 2019). Consistent with these findings, 500 nM guangxitoxin-1E effectively blocked Kv2.1_2.1 ($n_a =$ 3) currents (Fig. 6*A–C*). Furthermore, recent research has revealed that guangxitoxin-1E can inhibit currents

mediated by Kv2.1/6.4 heterotetrameric channels (Stewart et al., 2024). To further investigate the toxin's effects on Kv2.1/6.4 channels with fixed stoichiometry, dimers of these channels (Kv2.1_6.4) were expressed in oocytes, and the currents were recorded in the presence of 500 nM guangxitoxin-1E (Table 2). The total current amplitude was reduced when compared to currents recorded from Kv2.1_6.4 ($n_a =$ 4) channels in the absence of the toxin

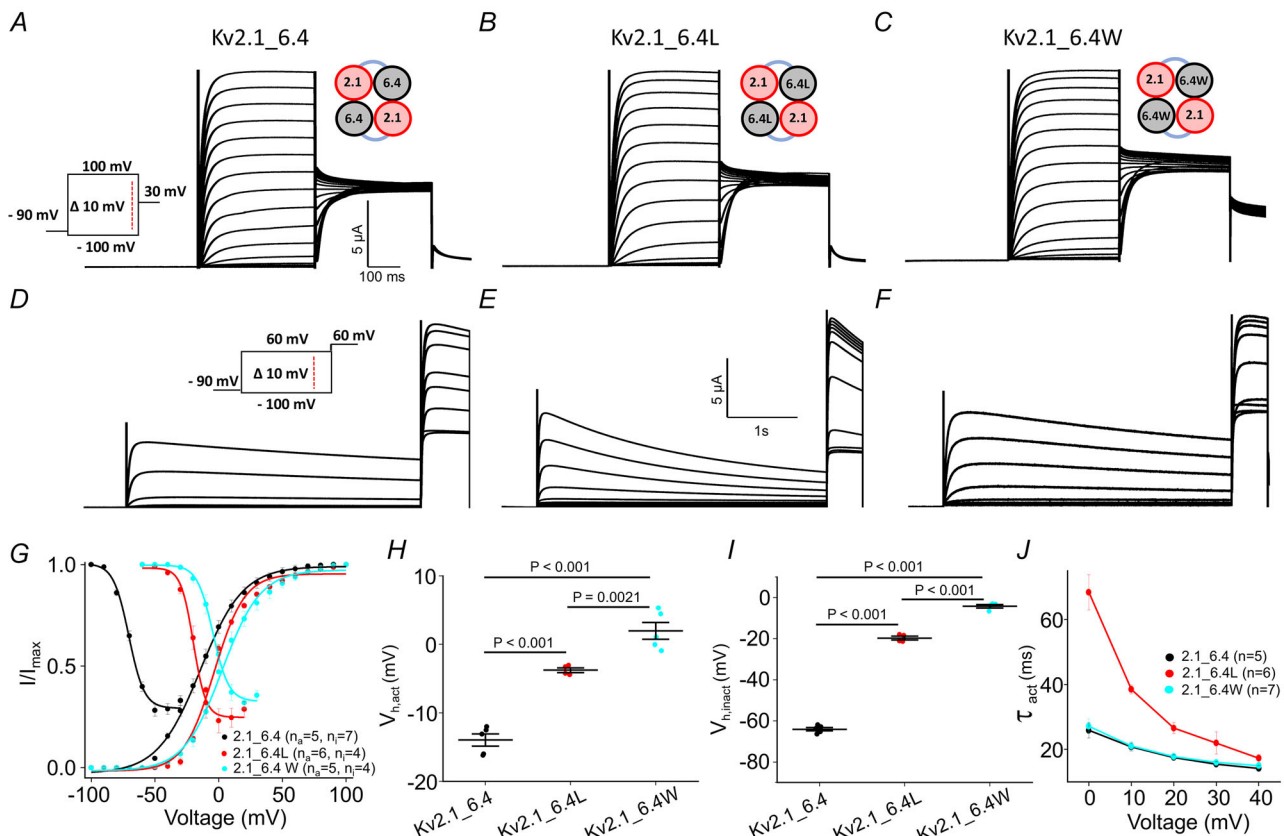

**Figure 5. Effect of mutations in the silent Kv6.4 subunits of Kv 2.1_6.4 concatemers**
*A–C*, representative current traces of activation with the cartoon of concatenated dimers (with leucine and tryptophan mutants at the charge transfer centre of the Kv6.4 subunit). *D–F*, example current traces for determining inactivation. *G–I*, steady-state activation and inactivation relationship of Kv 2.1_6.4, leucine and tryptophan mutant obtained from the tail currents. *J*, time constant of activation of Kv2.1_6.4 and the other two mutants. [Colour figure can be viewed at wileyonlinelibrary.com]

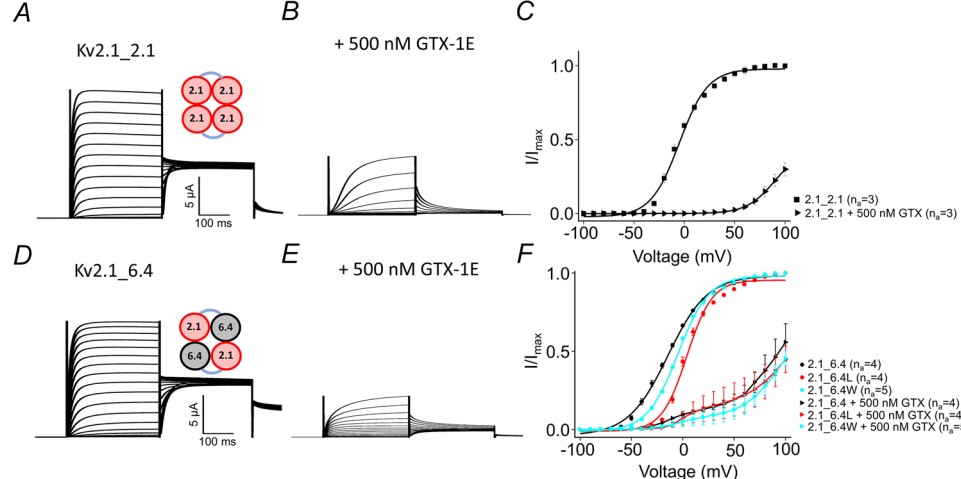

**Figure 6. Sensitivity of concatemers to guangxitoxin-1E (GTX-1E)**
*A*, *B*, *D* and *E*, representative current traces of Kv 2.1 monomers and the Kv 2.1_6.4 concatemer in the absence and presence of 500 nM GTX-1E. *C* and *F*, steady-state activation relationship of Kv2.1, Kv2.1_6.4, Kv2.1_6.4L and Kv2.1_6.4W mutants in the absence and presence of GTX-1E. [Colour figure can be viewed at wileyonlinelibrary.com]

(Fig. 6*D* and *E*). The $I/I_{max}$ relationship showed a distinct component in the presence of guangxitoxin-1E that was not observed in Kv2.1_2.1 dimeric channels under similar conditions. The blocking effect of GTX-1E is even stronger than the disabling effect of the leucine mutation in Kv2.1 subunits. This allowed us to further dissect the function of the different subunits in heteromeric Kv2.1/6.4 channels. The absence of any activation in Kv2.1 channels in the presence of 500 nM GTX-1E until voltages of 50 mV and the small remaining activation between 0 and +50 mV could indicate a range where partial activation by only two 6.4 subunits is sufficient to open the channel. A similar component was also observed for the leucine ($n_a$ = 4) and tryptophan ($n_a$ = 5) mutants (Fig. 6*F*).

### Role of phenylalanine F273 in the Kv6.4 subunit

The phenylalanine residue at the charge transfer centre has been identified as a critical determinant of Kv channel function by analysing the effects of mutations on the gating. In the 6.4 subunit, mutations to leucine and tryptophan residues exhibit less pronounced effects on channel activation but notably shift inactivation towards depolarizing potentials. We systematically substituted F273 with amino acids of varying charge (D ($n_a$ = 4, $n_i$ = 4), E ($n_a$ = 4, $n_i$ = 4), K ($n_a$ = 9, $n_i$ = 5), and R ($n_a$ = 5, $n_i$ = 4)), aliphatic (A ($n_a$ = 5, $n_i$ = 4), V ($n_a$ = 4, $n_i$ = 5), and IL ($n_a$ = 6, $n_i$ = 6)), and aromatic residues (Y ($n_a$ = 4, $n_i$ = 4)). The resulting activation and inactivation relationships are presented in Fig. 7. Most F273 mutations cause a significant depolarizing shift in steady-state inactivation while exerting lesser or even no impact on steady-state activation. This distinct behaviour suggests that phenylalanine plays a very special role in

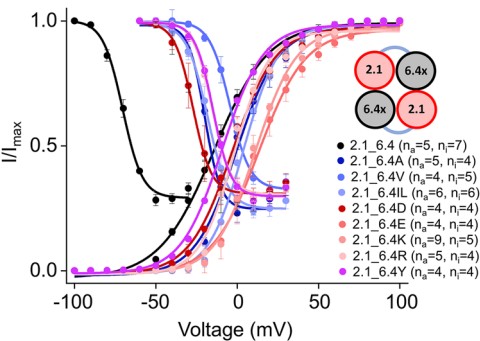

**Figure 7. Unique role of phenylalanine in the CTC of Kv6.4**
Cartoon displaying the tetrameric arrangement of two types of subunits, that is, 2.1 in red and 6.4*X* in black. The *X* represents different substitutions at the CTC of Kv 6.4. Normalized current ($I/I_{max}$) plotted against voltage (mV) for wild-type and different mutants. Wild-type (Kv2.1_6.4) is represented by the black curve. Mutants of the Kv6.4 subunit are represented by curves in different colours. Data points represent means ± SD, fitted to Eqn (1). [Colour figure can be viewed at wileyonlinelibrary.com]

stabilizing the charge transfer centre during inactivation, which cannot be fully compensated for by any other of the tested residues. These findings highlight the unique contribution of phenylalanine to the functional, and most likely also structural, integrity of the charge transfer centre in the silent Kv6.4 subunit of the Kv 2.1/6.4 heteromer that particularly controls the inactivation process.

### Recovery from open and closed state inactivation in Kv2.1_6.4 channel and mutants

The Kv2.1_6.4 channel exhibits a dissociation between the voltage dependencies of macroscopic activation and

inactivation that leads to a minimal overlap between the corresponding relationships. Previous studies have reported that in Kv4.2 channels these discrepancies are preferentially linked to closed-state inactivation (CSI). Kv6.4 has been shown to induce CSI when co-expressed with Kv2.1. We therefore studied the recovery from CSI in our Kv2.1_6.4 dimers by a protocol used previously (David et al., 2015) as shown in Fig. 8*A*. It began with a 100 ms control pulse at +60 mV to measure the initial current amplitude. This was followed by a 400 ms pulse at −130 mV to enable full recovery of all channels from inactivation. Next, a 5-s pulse at −60 mV was applied to induce closed-state inactivation. A pulse at −90 mV was then used to facilitate recovery from inactivation. Finally, a second test pulse to +60 mV was applied to assess the recovered current. The traces for Kv2.1_6.4,

Kv2.1_6.4L and Kv2.1_6.4W using this protocol are shown in Fig. 8*B–D*. The results revealed that the Kv2.1_6.4 channel's recovery rate of 244.4 ± 47.4 ms ($n = 6$) was consistent with our previous study. As shown in Fig. 8*G* the degree of CSI was approximately 50%. In contrast, the leucine (1167.1 ± 179.2 ms, $n = 6$) and tryptophan (574.5 ± 58.0 ms, $n = 6$) mutants displayed markedly slower recovery speed, accompanied by a lower degree of recovery from the CSI (Fig. 8*E–G*).

An appropriate protocol was used to study open-state inactivation (OSI) where instead of the pulse to −60 mV a pulse to +60 mV was applied to open the channels (Fig. 9*A–D*). The OSI results differed significantly from those of the CSI. Kv 2.1_6.4 and its corresponding mutants had a minimal impact on both the recovery time constant and the degree of OSI (Fig. 9*E–G*).

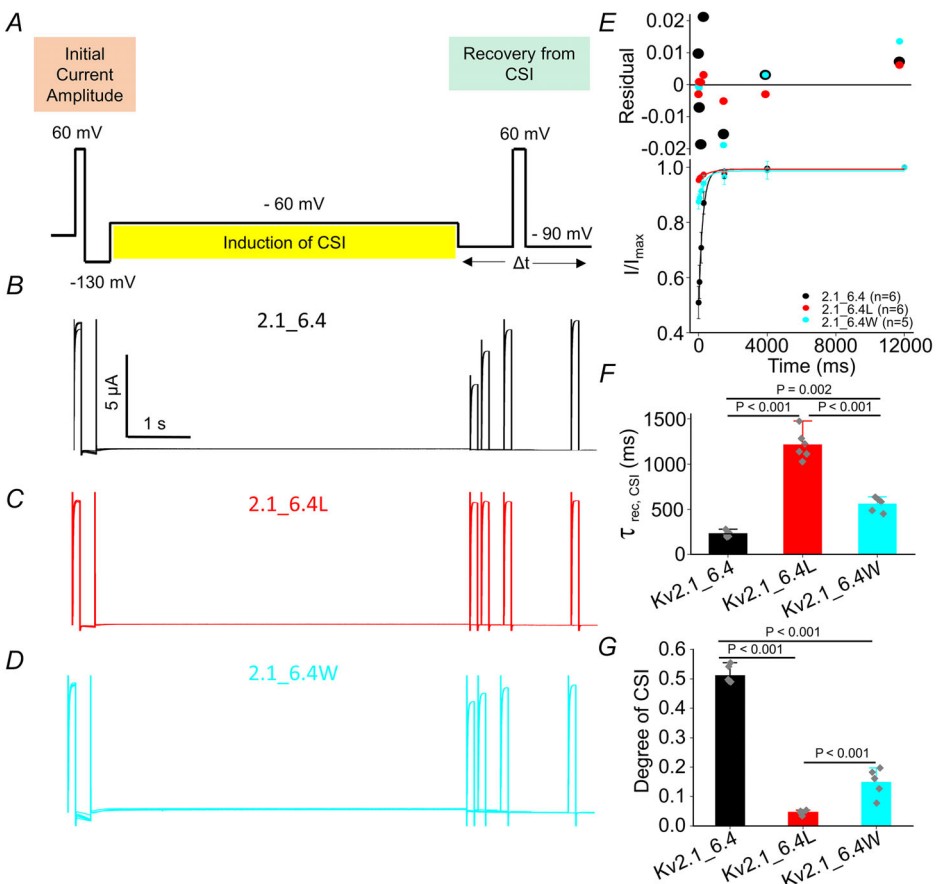

**Figure 8. Mutation at the CTC of the Kv6.4 subunit only in Kv2.1_6.4 channels slows the recovery from CSI**

*A–D*, representative current recordings were obtained using the voltage protocol shown at the top to study the recovery of Kv2.1_6.4 from CSI (details are provided in the Results section). *E*, recovery of Kv2.1_6.4 and the two mutants' leucine and tryptophan from CSI was analysed by plotting the normalized $I/I_{max}$ current amplitudes at +60 mV as a function of the pulse duration at the −90 mV voltage pulse. Single exponential functions were fitted to the data for the Kv2.1_6.4, Kv 2.1_6.4L and Kv 2.1_6.4W (black, red and cyan, respectively). The upper panel displays the residuals of the single exponential fits from CSI. *F*, recovery time constants $\tau_{rec,CSI}$ for the closed-state inactivation of Kv2.1_6.4 (black) Kv 2.1_6.4L (red) and Kv2.1_6.4W (cyan). *G*, degree of CSI for the three constructs was determined by subtracting the ratio of the initial current to the final current from one. [Colour figure can be viewed at wileyonlinelibrary.com]

## Discussion

The voltage-gated potassium channel Kv2.1 and its heteromeric complex with the silent subunits Kv6.4 represent an elegant model for studying the effect of this silent subunit on channel activation and inactivation. The findings of this study provide relevant insights into the function of the charge transfer centre of Kv6.4 (F273) on voltage-dependent activation, inactivation and recovery processes, and in particular, highlight the unique role of the phenylalanine residue, which even cannot be replaced by a tyrosine differing by only a hydroxyl group.

The introduction of leucine and tryptophan at the charge transfer centre of Kv6.4 (F273) produced distinct effects on the voltage-dependent properties of Kv2.1_Kv6.4 heteromeric channels. Both mutants exhibited depolarized $V_{h,act}$ values, consistent with reduced sensitivity to membrane depolarization during activation. The steady-state inactivation relationships of the mutants shifted significantly toward depolarized potentials, thus reducing the extent of inactivation. Hence, F273 seems to be an optimized constellation insofar that Kv2.1/6.4 channels are available in natural cells at more negative potentials, possibly contributing to stabilizing the resting potential. F273 also appears uniquely suited to stabilizing the process of repolarization in a nerve action potential. The hallmark of Kv6.4 incorporation into heteromeric Kv2.1/6.4 channels, the pronounced inactivation, is strongly affected by mutating the phenylalanine in the CTC of the Kv6.4 subunit.

The recovery kinetics from closed-state inactivation (CSI) were significantly altered by F273 mutations in Kv6.4. The leucine and tryptophan mutants displayed

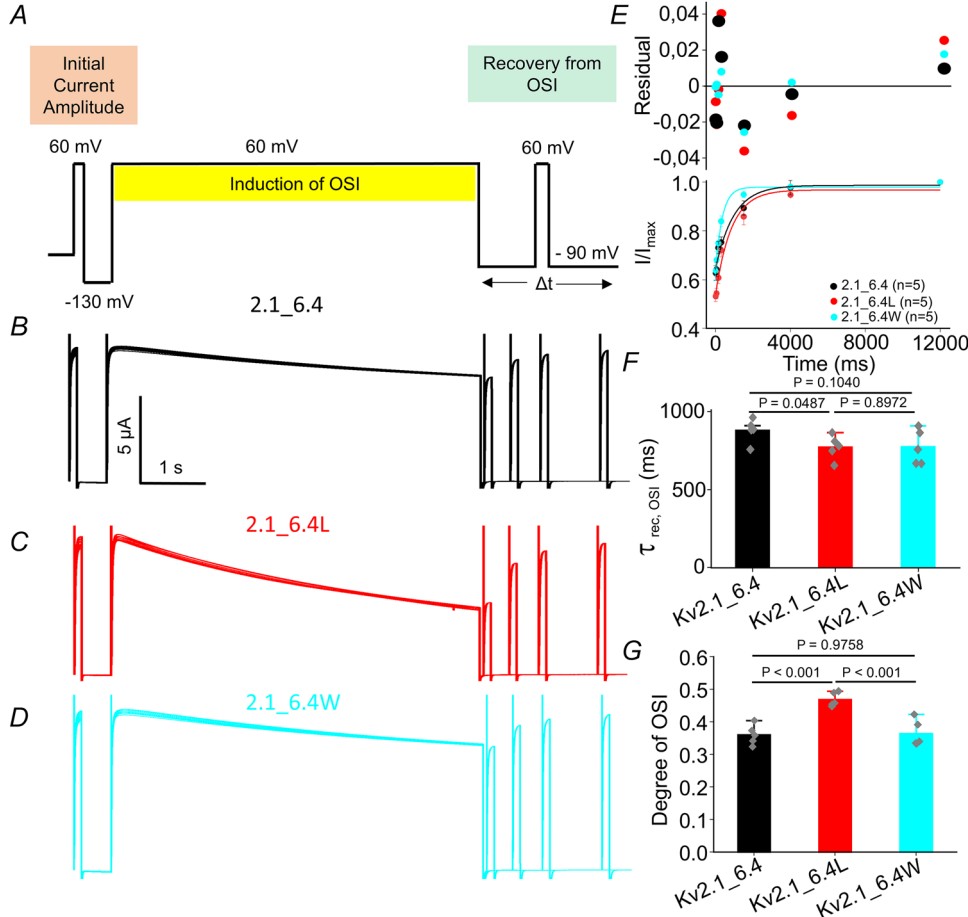

**Figure 9. Effects of mutations at the CTC of the Kv6.4 subunit on the recovery from OSI**
*A*, experimental protocol. Currents were measured by depolarizing to 60 mV, followed by a prolonged depolarization to induce OSI. Recovery from OSI was assessed by applying a test pulse after varying recovery intervals at −90 mV. *B–D*, representative current traces showing induction and recovery from OSI. *E*, time course of recovery from OSI, presented as normalized current amplitude ($I/I_{max}$) plotted against recovery time for wild-type (Kv2.1_6.4) and mutant channels (Kv2.1_6.4L), Kv2.1_6.4W). Residuals of single exponential fits are shown in the upper panel at extended ordinate. Data points represent means ± SD. *F*, recovery time constant ($\tau_{rec,OSI}$) for each channel variant, indicating no significant difference between wild-type and mutants. *G*, degree of OSI measured as the fraction of inactivated current. [Colour figure can be viewed at wileyonlinelibrary.com]

markedly slower recovery rates compared to the wild-type Kv2.1_6.4 channel, with prolonged recovery time. In contrast, the degree of closed-state inactivation is reduced for the two mutants. This suggests that the induction of inactivation must be faster for the wild-type. The fast induction of inactivation and fast recovery from it in the wild-type Kv2.1_6.4 channel could be explained by a high degree of flexibility or a lower energy barrier in the CTC when the conserved phenylalanine is positioned there. Interestingly, recovery from OSI remained largely unaffected by these mutations, indicating that the functional role of F273 is more critical in the context of CSI. This divergence underscores the complexity of gating mechanisms in heteromeric channels and suggests that distinct structural determinants govern recovery from closed *versus* open states.

Our results can also be illustrated by a simple state model containing two closed states, one open state and two inactivated states (Fig. 10). Activation is reduced to a two-state process along C–C–O (blue arrows). Inactivation can take place from the second C to $I_C$ (CSI) and from O to $I_O$ (OSI). The colours of the arrows in Fig. 10 correspond to those in the other figures (Kv2.1_6.4, black; Kv2.1_6.4L, red; Kv2.1_6.4W, cyan). For the L and

W mutation, CSI is strongly attenuated whereas OSI is moderate for all WT and both mutants. The recovery from CSI is much faster in WT than in both mutants whereas the recovery from OSI is similarly slow for all constructs, suggesting that opening drives the WT channel to more extensive conformational changes in the inactivated state which occludes a fast recovery. This process would also be occluded by both mutants for the recovery from the CSI. This in turn suggests, that the energy barrier to run into $I_C$ must be lower in WT than in the two mutants.

Mutations in the Kv6.4 subunit are linked to specific diseases like familial migraine and labour pain (Lacroix et al., 2024; Lafrenière & Rouleau, 2012; Lee et al., 2020). Understanding the biophysics of the silent Kv6.4 sub-unit in heterotetrameric Kv2.1/6.4 channels is crucial for advancing knowledge of these channels and potentially developing targeted medical treatments.

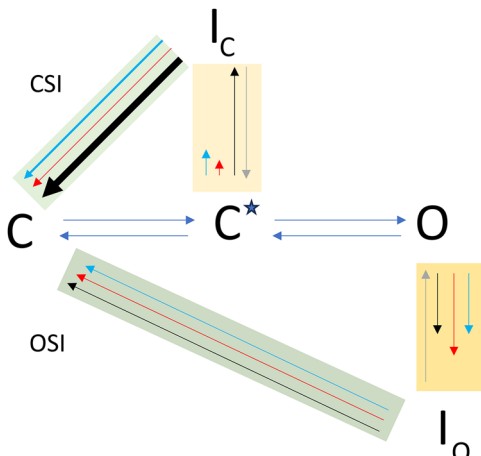

**Figure 10. Gating scheme for Kv2.1_6.4 channel**
A simplified state model with two closed states, one open state and two inactivated states to be used to interpret our data. Colour coded arrows: equal voltage-dependent activation, blue. The degree of inactivation is given by the two C–I equilibria. Different inactivation for constructs: Kv2.1_6.4, black; Kv2.1_6.4L, red; Kv2.1_6.4W, cyan. Thick arrows indicate fast, thin arrows, slow processes. Yellow light background: CSI: WT 50%; L and W mutation: inactivation strongly attenuated. Yellow dark background: OSI: all moderate, L mutation some larger. These arrows indicate rates. For the recovery from inactivation (green background) a time constant is considered. Thick arrows: fast; thin arrows: slow. See also text in the discussion. So, a conclusion is that CSI is extended in WT and that the two mutations strongly weaken it. It is very interesting that the extended CSI in WT can recover more quickly than in both mutants. Hence, the energy barrier to run into $I_C$ must be lower in WT than in the two mutations. [Colour figure can be viewed at wileyonlinelibrary.com]

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

## Additional information

### Data availability statement

All the data in the present study are available from the corresponding author upon reasonable request.

### Competing interests

The authors declare that they have no known competing financial interests or personal relationships that could have appeared to influence the work reported in this paper.

### Author contributions

D.T. did the measurements, analysed the data, and wrote the manuscript. C.S. did the molecular biology and edited

the manuscript. K.B. supervised the study and edited the manuscript. All authors have read and approved the final version of this manuscript and agree to be accountable for all aspects of the work in ensuring that questions related to the accuracy or integrity of any part of the work are appropriately investigated and resolved. All persons designated as authors qualify for authorship, and all those who qualify for authorship are listed.

## Funding

The work was supported by the Jena University Hospital.

## Acknowledgements

We thank U. Enke, S. Bernhardt, C. Ranke and U. Singer for excellent technical assistance.

## Keywords

charge transfer centre, closed state inactivation, concatemers, guangxitoxin, Kv2.1/6.4

## Supporting information

Additional supporting information can be found online in the Supporting Information section at the end of the HTML view of the article. Supporting information files available:

**Peer Review History**
**Supporting Information**

