## [Peer Review History · The Journal of Physiology]

Functional control of heteromeric Kv2.1/6.4 channels by the voltage sensor domain of the silent Kv6.4 subunit

Debanjan Tewari, Christian Sattler, and Klaus Benndorf
DOI: 10.1113/JP288376

Corresponding author(s): Debanjan Tewari (Debanjan.Tewari@med.uni-jena.de)

The following individual(s) involved in review of this submission have agreed to reveal their identity: Jon Sack (Referee #1)

Review Timeline:

Submission Date:	19-Dec-2024
Editorial Decision:	04-Feb-2025
Revision Received:	03-Mar-2025
Editorial Decision:	26-Mar-2025
Revision Received:	04-Apr-2025
Editorial Decision:	25-Apr-2025
Revision Received:	30-Apr-2025
Accepted:	12-May-2025

Senior Editor: Peking Fong

Reviewing Editor: Brian Delisle

Transaction Report:

Dear Dr Tewari,

Re: JP-RP-2024-288376 "Functional control of heteromeric Kv2.1/6.4 channels by the voltage sensor domain of the silent Kv6.4 subunit" by Debanjan Tewari, Christian Sattler, and Klaus Benndorf

Thank you for submitting your manuscript to The Journal of Physiology. It has been assessed by a Reviewing Editor and by 2 expert referees and we are pleased to tell you that it is potentially acceptable for publication following satisfactory major revision.

REVISION CHECKLIST:

Please upload two versions of your manuscript text: one with all relevant changes highlighted and one clean version with no

changes tracked. The manuscript file should include all tables and figure legends, but each figure/graph should be uploaded as separate, high-resolution files.

We look forward to receiving your revised submission.

Yours sincerely,

Peying Fong
Senior Editor
The Journal of Physiology

REQUIRED ITEMS

- Author photo and profile. First or joint first authors are asked to provide a short biography (no more than 100 words for one author or 150 words in total for joint first authors) and a portrait photograph. These should be uploaded and clearly labelled together in a Word document with the revised version of the manuscript. See Information for Authors for further details.
- Your manuscript must include a complete Additional Information section, including competing interests; funding; author contributions and acknowledgements.
- The Journal of Physiology funds authors of provisionally accepted papers to use the premium BioRender site to create high resolution schematic figures. Follow this link and enter your details and the manuscript number to create and download figures. Upload these as the figure files for your revised submission. If you choose not to take up this offer, we require figures to be of similar quality and resolution. If you are opting out of this service to authors, state this in the Comments section on the Detailed Information page of the submission form. The link provided should only be used for the purposes of this submission. Authors will be charged for figures created on this premium BioRender account if they are not related to this manuscript submission.
- Please upload separate high-quality figure files via the submission form.
- Please ensure that the Article File you upload is a Word file.
- Papers must comply with the Statistics Policy: https://jp.msubmit.net/cgi-bin/main.plex?form_type=display_requirements#statistics.

In summary:

- If n {less than or equal to} 30, all data points must be plotted in the figure in a way that reveals their range and distribution. A bar graph with data points overlaid, a box and whisker plot or a violin plot (preferably with data points included) are acceptable formats.
- If $n > 30$, then the entire raw dataset must be made available either as supporting information, or hosted on a not-for-profit repository, e.g. FigShare, with access details provided in the manuscript.
- 'n' clearly defined (e.g. x cells from y slices in z animals) in the Methods. Authors should be mindful of pseudoreplication.
- All relevant 'n' values must be clearly stated in the main text, figures and tables.
- The most appropriate summary statistic (e.g. mean or median and standard deviation) must be used. Standard Error of the Mean (SEM) alone is not permitted.
- Exact p values must be stated. Authors must not use 'greater than' or 'less than'. Exact p values must be stated to three significant figures even when 'no statistical significance' is claimed.

- A Data Availability Statement is required for all papers reporting original data. This must be in the Additional Information section of the manuscript itself. It must have the paragraph heading 'Data Availability Statement'. All data supporting the results in the paper must be either: in the paper itself; uploaded as Supporting Information for Online Publication; or archived in an appropriate public repository. The statement needs to describe the availability or the absence of shared data. Authors must include in their statement: a link to the repository they have used, or a statement that it is available as Supporting Information; reference the data in the appropriate section(s) of their manuscript; and cite the data they have shared in the References section. Whenever possible, the scripts and other artefacts used to generate the analyses presented in the paper should also be publicly archived. If sharing data compromises ethical standards or legal requirements then authors are not expected to share it, but must note this in their statement. For more information, see our Statistics Policy.

- Please include an Abstract Figure file, as well as the Figure Legend text within the main article file. The Abstract Figure is a piece of artwork designed to give readers an immediate understanding of the research and should summarise the main conclusions. If possible, the image should be easily 'readable' from left to right or top to bottom. It should show the physiological relevance of the manuscript so readers can assess the importance and content of its findings. Abstract Figures should not merely recapitulate other figures in the manuscript. Please try to keep the diagram as simple as possible and without superfluous information that may distract from the main conclusion(s). Abstract Figures must be provided by authors no later than the revised manuscript stage and should be uploaded as a separate file during online submission labelled as File Type 'Abstract Figure'. Please also ensure that you include the figure legend in the main article file. All Abstract Figures should be created using BioRender. Authors should use The Journal's premium BioRender account to export high-resolution images. Details on how to use and access the premium account are included as part of this email.

Reviewing Editor's comments:

Two experts in the field have evaluated the manuscript. Overall, referee 1 found the work of interest in the field but identified several significant concerns that limited its impact on the field. Referee 2 identified several major problems with the manuscript and the data description that also limited their enthusiasm for the work.

Senior Editor:

Comments to ensure the paper complies with the Statistics Policy:

Please ensure compliance with the journal's statistics policy; please provide standard deviations and not standard errors of the mean, as well as exact p values for analyses involving sample sizes fewer than 30, as described in the policy. Please also note policy regarding supplementary figures, which must be included in the manuscript proper.

Comments to the authors:

Initial review of your manuscript, "Functional control of heteromeric Kv2.1/6.4 channels by the voltage sensor domain of the silent Kv6.4 subunit" is now complete.

Attached you will find detailed reviews from both Expert Referees, accompanied by a summary from the Reviewing Editor. While enthusiasm for the study's potential impact is voiced by Referee #1, this referee's specific points do suggest that several conclusions require greater scrutiny and more considered interpretation. Referee #2 felt that numerous lapses in data presentation precluded rigorous assessment of the study, of major concern. Overall, concerns shared by both Expert Referees pertain to lapses in presentation and narration of the results.

Nonetheless, this topic is of sufficient potential impact to the collective knowledge base that an opportunity to address the points raised in initial review should be offered. Based on the nature of the comments received, this will entail a major revision. Please address all comments in your revised manuscript. In addition, please ensure compliance with The Journal of Physiology's published Statistical Policies. Note also that figures such as those included in the present version as Supplemental Figures instead should be incorporated fully within the manuscript proper.

I hope that you are open to this possibility, and look forward to receiving your revised manuscript.

Referee #1:

This manuscript presents an elegant investigation into the role of the Kv6.4 subunit in heterotetrameric Kv2.1/6.4 channels. The study offers compelling perturbative experiments to address ideas based on voltage region correlations regarding the voltage-dependent inactivation and voltage sensor contributions of Kv6.4. This work represents a significant advance in our understanding of these channels. The authors employ concatenated dimers of Kv2.1 and Kv6.4 subunits in a strategy designed to minimize assembly of Kv2.1 homomers. An insight from this work is the distinction between the contributions of

Kv6.4 to closed-state inactivation versus open-state inactivation. The authors convincingly demonstrate that Kv6.4 selectively influences the former while sparing the latter, illustrating that these are separable processes. This finding provides a significant conceptual advance in the understanding of Kv channel inactivation dynamics. The study investigates the role of the conserved phenylalanine (F273) in the S2 helix of the Kv6.4 subunit, uncovering its distinct impact on steady-state inactivation and recovery kinetics. The shift in inactivation voltage and the slowed recovery from closed-state inactivation upon F273 mutation underscore the unique functional role of this residue. These results offer mechanistic depth and advance the field's understanding of how silent subunits modulate channel behavior, providing a framework for understanding silent subunit contributions in heteromeric channels more broadly. The manuscript is clearly presented and well-written. I have only a few suggestions, which are minor, yet important to address.

Specific Suggestions:

1. Line 292: "These results indicate ... that the channels built from monomers assemble in the 2:2 ratio, with Kv6.4 subunits opposite to each other."
 - o I disagree that this conclusion is justified from this particular data: there are plenty of examples of functional concatenated encoding sequences not leading to the intended protein architecture.
2. Line 295: "... the dimer exhibited improved expression compared to expression of the monomers."
 - o Data missing. Please quantify conductance densities of Kv6.4-2.1 and Kv2.1-6.4 dimers vs Kv2.1 + Kv6.4 co-injections.
3. Line 312: "The I/I_{max} relationship showed a distinct component in the presence of guangxitoxin-1E that was not observed in Kv2.1 homomeric channels under similar conditions. This extra component suggests that the assumed two Kv6.4 subunits can partly activate the Kv2.1_6.4 channels on their own if the Kv2.1 subunits are blocked."
 - o It's also possible that Kv6.4 worsens the GTX binding affinity. As the extra conductance components in GTX have a $V_{h,act}$ similar to channels without GTX, could it be that they are simply apo channels not bound to GTX? Extra component looks similar to GV of 10 nM GTX in Fig 1 of PMID: 30397012. Testing a higher concentration of GTX could settle this.
4. Line 344: "In contrast, the leucine (3846.9 {plus minus} 212.5 ms) and tryptophan (1442.5 {plus minus} 85.6 ms) mutants displayed markedly slower recovery speed, accompanied by a lower degree of recovery from the CSI."
 - o Is it possible that the slow recovery from the CSI protocol for Kv 2.1_6.4L and Kv 2.1_6.4W is recovery from the same state as induced by the OSI protocol?
 - o To my eye, the recovery looks multiphasic and some of the fits in Fig 5C and 6C look more complex than single exponentials. Please show actual representative single exponential fits if that is not what is shown. What are the residuals on the fits?

Referee #2:

The role of the charge transfer center (a Phenylalanine residue in the channel S2 segment that influences movement in the charged Arginine residues in the S4 domain) is an interesting and important topic in voltage-activated ion channels. Here the authors investigate the role of the CTC by making mutations at F in Kv2.1 and KvS (silent) subunits and by generating dimers of Kv2.1 and KvS subunits to identify subunit contributions to gating.

I found that the results of this work difficult to evaluate because the text in the Results section did not describe the experimental results shown in the Figures. For instance, in the first few paragraphs of the Results section, the authors refer to Figure 1 only when describing the voltage pulse protocol and not the experimental results. Unfortunately this trend

continues for most of the manuscript. I would suggest that the authors rewrite the results section to describe the results presented in each panel of each of the Figures.

The introduction could also be shortened as could the discussion.

END OF COMMENTS

Dear Dr Tewari,

Re: JP-RP-2024-288376 "Functional control of heteromeric Kv2.1/6.4 channels by the voltage sensor domain of the silent Kv6.4 subunit" by Debanjan Tewari, Christian Sattler, and Klaus Benndorf

Thank you for submitting your manuscript to The Journal of Physiology. It has been assessed by a Reviewing Editor and by 2 expert referees and we are pleased to tell you that it is potentially acceptable for publication following satisfactory major revision.

Your revised manuscript should be submitted online using the link in your Author Tasks: <https://jp.msubmit.net/cgi-bin/main.plex?el=A3JS3GzS4A3hQQ1F5A9ftdhrhVOZBnJYjJ0sw2nx6UiQZ>. This link is accessible via your account as Corresponding Author; it is not available to your co-authors. If this presents a problem, please contact journal staff (jp@physoc.org). Image files from the previous version are retained on the system. Please ensure you replace or remove any files that are being revised.

LANGUAGE EDITING AND SUPPORT FOR PUBLICATION: If you would like help with English language editing, or other article preparation support, Wiley Editing Services offers expert help, including English Language Editing, as well as translation, manuscript formatting, and figure formatting at www.wileyauthors.com/eoo/preparation. You can also find resources for Preparing Your Article for general guidance about writing and preparing your manuscript at www.wileyauthors.com/eoo/prepresources.

REVISION CHECKLIST:

We look forward to receiving your revised submission.

Yours sincerely,

Peying Fong
Senior Editor
The Journal of Physiology

REQUIRED ITEMS

- Author photo and profile. First or joint first authors are asked to provide a short biography (no more than 100 words for one author or 150 words in total for joint first authors) and a portrait photograph. These should be uploaded and clearly labelled together in a Word document with the revised version of the manuscript. See Information for Authors for further details.

Authors photo and Profile has been provided.

- Your manuscript must include a complete Additional Information section, including competing interests; funding; author contributions and acknowledgements.

We would like to express our gratitude to the senior editor. The revised version of the manuscript now includes additional information sections covering competing interests, funding, author contributions, and acknowledgments.

- The Journal of Physiology funds authors of provisionally accepted papers to use the premium BioRender site to create high resolution schematic figures. Follow this link and enter your details and the manuscript number to create and download figures. Upload these as the figure files for your revised submission. If you choose not

to take up this offer, we require figures to be of similar quality and resolution. If you are opting out of this service to authors, state this in the Comments section on the Detailed Information page of the submission form. The link provided should only be used for the purposes of this submission. Authors will be charged for figures created on this premium BioRender account if they are not related to this manuscript submission.

We have submitted a high-resolution version of the figures.

- Please upload separate high-quality figure files via the submission form.

High quality figures have been uploaded as a separate file.

- Please ensure that the Article File you upload is a Word file.

We have uploaded the Word File as instructed.

- Papers must comply with the Statistics Policy: https://jp.msubmit.net/cgi-bin/main.plex?form_type=display_requirements#statistics.

In summary:

- If n {less than or equal to} 30, all data points must be plotted in the figure in a way that reveals their range and distribution. A bar graph with data points overlaid, a box and whisker plot or a violin plot (preferably with data points included) are acceptable formats.

- If $n > 30$, then the entire raw dataset must be made available either as supporting information, or hosted on a not-for-profit repository, e.g. FigShare, with access details provided in the manuscript.

- ' n ' clearly defined (e.g. x cells from y slices in z animals) in the Methods. Authors should be mindful of pseudoreplication.

- All relevant ' n ' values must be clearly stated in the main text, figures and tables.

- The most appropriate summary statistic (e.g. mean or median and standard deviation) must be used. Standard Error of the Mean (SEM) alone is not permitted.

- Exact p values must be stated. Authors must not use 'greater than' or 'less than'. Exact p values must be stated to three significant figures even when 'no statistical significance' is claimed.

The data in the manuscript are now at par with the journal's statistical policy. Individual data sets are presented as box plots, with ' n ' values clearly indicated in the figures, text, and tables. Results in the figures and tables are reported as mean values with standard deviations. Exact p -values are provided in the figures, and the raw data have been uploaded as supporting information.

- A Data Availability Statement is required for all papers reporting original data. This must be in the Additional Information section of the manuscript itself. It must have the paragraph heading 'Data Availability Statement'. All data supporting the results in the paper must be either: in the paper itself; uploaded as Supporting Information for Online Publication; or archived in an appropriate public repository. The statement needs to describe the availability or the absence of shared data. Authors must include in their statement: a link to the repository they have used, or a statement that it is available as Supporting Information; reference the data in the appropriate sections(s) of their manuscript; and cite the data they have shared in the References section. Whenever possible, the scripts and other artefacts used to generate the analyses presented in the paper should also be publicly archived. If sharing data compromises ethical standards or legal requirements then authors are not expected to share it, but must note this in their statement. For more information, see our Statistics Policy.

A data availability statement has been incorporated into the main manuscript.

- Please include an Abstract Figure file, as well as the Figure Legend text within the main article file. The Abstract Figure is a piece of artwork designed to give readers an immediate understanding of the research and should summarise the main conclusions. If possible, the image should be easily 'readable' from left to right or top to bottom. It should show the physiological relevance of the manuscript so readers can assess the importance and content of its findings. Abstract Figures should not merely recapitulate other figures in the manuscript. Please try to keep the diagram as simple as possible and without superfluous information that may distract from the main conclusion(s). Abstract Figures must be provided by authors no later than the revised manuscript stage and should be uploaded as a separate file during online submission labelled as File Type 'Abstract Figure'. Please also ensure that you include the figure legend in the main article file. All Abstract Figures should be created using BioRender. Authors should use The Journal's premium BioRender account to export high-resolution images. Details on how to use and access the premium account are included as part of this email.

The abstract figure file has been created, and the figure legend text has been incorporated into the main article file.

We sincerely appreciate the reviewers' interest and thoughtful comments on our work. In our revised manuscript, we have incorporated all the suggestions and critiques from the two referees, as well as the feedback from the Reviewing Editor and Senior Editor. All the response are highlighted in red. Our revisions address all comments and further strengthen our main conclusions. Below, we provide detailed responses to each referee's concerns.

Reviewing Editor's comments:

Two experts in the field have evaluated the manuscript. Overall, referee 1 found the work of interest in the field but identified several significant concerns that limited its impact on the field. Referee 2 identified several major problems with the manuscript and the data description that also limited their enthusiasm for the work.

Senior Editor:

Comments to ensure the paper complies with the Statistics Policy:

Please ensure compliance with the journal's statistics policy; please provide standard deviations and not standard errors of the mean, as well as exact p values for analyses involving sample sizes fewer than 30, as described in the policy. Please also note policy regarding supplementary figures, which must be included in the manuscript proper.

As previously mentioned, the data in the manuscript now aligns with the journal's statistical policy. Individual datasets are presented as box plots, with 'n' values clearly indicated in the figures, text, and tables. The results shown in the figures and tables represent mean values with standard deviations, and exact p-values are provided in the figures. Additionally, the supplementary figure has been incorporated into the manuscript as Figures 2 and 3.

Comments to the authors:

Initial review of your manuscript, "Functional control of heteromeric Kv2.1/6.4 channels by the voltage sensor domain of the silent Kv6.4 subunit" is now complete.

Attached you will find detailed reviews from both Expert Referees, accompanied by a summary from the Reviewing Editor. While enthusiasm for the study's potential impact is voiced by Referee #1, this referee's specific points do suggest that several conclusions require greater scrutiny and more considered interpretation. Referee #2 felt that numerous lapses in data presentation precluded rigorous assessment of the study, of major concern. Overall, concerns shared by both Expert Referees pertain to lapses in presentation and narration of the results.

Nonetheless, this topic is of sufficient potential impact to the collective knowledge base that an opportunity to address the points raised in initial review should be offered. Based on the nature of the comments received, this will entail a major revision. Please address all comments in your revised manuscript. In addition, please ensure compliance with The Journal of Physiology's published Statistical Policies. Note also that figures such as those included in the present version as Supplemental Figures instead should be incorporated fully within the manuscript proper.

I hope that you are open to this possibility, and look forward to receiving your revised manuscript.

Referee #1:

This manuscript presents an elegant investigation into the role of the Kv6.4 subunit in heterotetrameric Kv2.1/6.4 channels. The study offers compelling perturbative experiments to address ideas based on voltage region correlations regarding the voltage-dependent inactivation and voltage sensor contributions of Kv6.4. This work represents a significant advance in our understanding of these channels. The authors employ concatenated dimers of Kv2.1 and Kv6.4 subunits in a strategy designed to minimize assembly of Kv2.1 homomers. An insight from this work is the distinction between the contributions of Kv6.4 to closed-state inactivation versus open-state inactivation. The authors convincingly demonstrate that Kv6.4 selectively influences the former while sparing the latter, illustrating that these are separable processes. This finding provides a significant conceptual advance in the understanding of Kv channel inactivation dynamics. The study investigates the role of the conserved phenylalanine (F273) in the S2 helix of the Kv6.4 subunit, uncovering its distinct impact on steady-state inactivation and recovery kinetics. The shift in inactivation voltage and the slowed recovery from closed-state inactivation upon F273 mutation underscore the unique functional role of this residue. These results offer mechanistic depth and advance the field's understanding of how silent subunits modulate channel behavior, providing a framework for understanding silent subunit contributions in heteromeric channels more broadly. The manuscript is clearly presented and well-written. I have only a few suggestions, which are minor, yet important to address.

Specific Suggestions:

1. Line 292: "These results indicate ... that the channels built from monomers assemble in the 2:2 ratio, with Kv6.4 subunits opposite to each other."

o I disagree that this conclusion is justified from this particular data: there are plenty of examples of functional concatenated encoding sequences not leading to the intended protein architecture.

We sincerely appreciate the reviewer for raising this important question and understand the concern. Our initial conclusion was based on electrophysiological data obtained from both monomeric and dimeric constructs. Specifically, when comparing the slope factor and V_{half} values between the two, we observed a high degree of similarity. This observation, along with previous findings reported in the literature (PMID: 32284408), led us to

our initial interpretation. However, upon further reflection and in response to the reviewer's feedback, we have decided to remove this section from our revised manuscript to ensure clarity and avoid any potential misinterpretation.

2. Line 295: "... the dimer exhibited improved expression compared to expression of the monomers."

o Data missing. Please quantify conductance densities of Kv6.4-2.1 and Kv2.1-6.4 dimers vs Kv2.1 + Kv6.4 co-injections.

We appreciate the reviewer's suggestion. However, we would like to clarify that we used TEVC for the measurements, making it not possible to quantify seriously the conductance densities. Instead, we have plotted the current at 100 mV for the three constructs—Kv 2.1_6.4, Kv 6.4_2.1, and Kv 2.1 6.4 (monomer)—from different oocytes. The data are presented below.

3. Line 312: "The I/Imax relationship showed a distinct component in the presence of guangxitoxin-1E that was not observed in Kv2.1 homomeric channels under similar conditions. This extra component suggests that the assumed two Kv6.4 subunits can partly activate the Kv2.1_6.4 channels on their own if the Kv2.1 subunits are blocked."

o It's also possible that Kv6.4 worsens the GTX binding affinity. As the extra conductance components in GTX have a $V_{h,act}$ similar to channels without GTX, could it be that they are simply apo channels not bound to GTX? Extra component looks similar to GV of 10 nM GTX in Fig 1 of PMID: 30397012. Testing a higher concentration of GTX could settle this.

We acknowledge the reviewer's concern. Kv6.4 is likely to diminish the binding of GTX-1E in the Kv2.1_6.4 concatenated channel. GTX-1E is a gating modifier that binds specifically to the voltage sensor region of the channel, rather than the pore domain. Kv2.1 is a homomeric, domain-swapped channel, and its interacting amino acids with GTX-1E are well characterized. In a single Kv2.1 channel, four toxin molecules can interact with four voltage sensor domains (VSDs), thereby interfering with channel opening. However, in the heteromeric concatenated dimer, there are two VSDs from Kv2.1 and two from Kv6.4. Since the VSDs of Kv6.4 lack the necessary interacting amino acids for GTX-1E binding, the toxin can only associate with the two VSDs of Kv2.1. This is clearly reflected in the I/Imax curve, which shows two components, indicating reduced toxin binding in the heteromeric condition compared to homomeric Kv2.1. Additionally, the probability of an apo form is significantly lower, as supported by the observed decrease in current (now shown in Figure 5). We believe that even at higher toxin concentrations, the contribution from Kv6.4 in the concatenated channel would remain unresolved.

4.0 Line 344: "In contrast, the leucine (3846.9 {plus minus} 212.5 ms) and tryptophan (1442.5 {plus minus} 85.6 ms) mutants displayed markedly slower recovery speed, accompanied by a lower degree of recovery from the CSI."

o Is it possible that the slow recovery from the CSI protocol for Kv 2.1_6.4L and Kv 2.1_6.4W is recovery from the same state as induced by the OSI protocol?

The reviewer is correct. It is possible that the slow recovery of the Leucine and Tryptophan mutants from the CSI is due to their recovery from the same state induced by the OSI protocol.

o To my eye, the recovery looks multiphasic and some of the fits in Fig 5C and 6C look more complex than single exponentials. Please show actual representative single exponential fits if that is not what is shown. What are the residuals on the fits?

We appreciate the reviewer's concern regarding the fit for the recovery experiment. We have carefully cross-checked the fits and would like to clarify that the recovery from CSI and OSI is not multiphasic. The fits presented in Figures 5C and 6C (now Figures 7C and 8C) are actual single-exponential fits. Additionally, we have provided the residuals of the fits below for the three constructs—Kv2.1_6.4, Kv2.1_6.4L, and Kv2.1_6.4W—with respect to CSI and OSI.

Referee #2:

The role of the charge transfer center (a Phenylalanine residue in the channel S2 segment that influences movement in the charged Arginine residues in the S4 domain) is an interesting and important topic in voltage-activated ion channels. Here the authors investigate the role of the CTC by making mutations at F in Kv2.1 and KvS (silent) subunits and by generating dimers of Kv2.1 and KvS subunits to identify subunit contributions to gating.

I found that the results of this work difficult to evaluate because the text in the Results section did not describe the experimental results shown in the Figures. For instance, in the first few paragraphs of the Results section, the authors refer to Figure 1 only when describing the voltage pulse protocol and not the experimental results. Unfortunately, this trend continues for most of the manuscript. I would suggest that the authors rewrite the results section to describe the results presented in each panel of each of the Figures.

We are thankful for the reviewer's valuable feedback and concerns regarding the formatting of our manuscript. In response, we have meticulously reviewed the document and made the necessary corrections to ensure consistency, clarity, and adherence. These revisions enhance the overall readability and presentation of our work.

The introduction could also be shortened as could the discussion.

We sincerely appreciate the reviewer's suggestion. In the revised manuscript, we have already shortened both the introduction and discussion to enhance clarity. We believe these revisions improve the overall readability and focus of the manuscript. In the introduction section, lines 130 to 137 have been removed, and in the results section, lines 365 to 370 have been revised.

END OF COMMENTS

The Physiological Society is a company limited by guarantee. Registered in England and Wales, No. 00323575. Registered Office: Hodgkin Huxley House, 30 Farringdon Lane, London, EC1R 3AW, UK. Registered Charity No. 211585. The Physiological Society and The Journal of Physiology are registered trademarks.

This email and any files transmitted with it are confidential and intended solely for the use of the individual or entity to whom they are addressed. If you have received this email in error please notify the sender. If you are not the named addressee you should not disseminate, distribute or copy this e-mail. The Physiological Society may monitor email traffic data.

The Physiological Society has taken reasonable precautions to ensure no viruses are present in this email, however does not accept responsibility for any loss or damage arising from the use of this email or attachments.

Dear Dr Tewari,

Re: JP-RP-2025-288376R1 "Functional control of heteromeric Kv2.1/6.4 channels by the voltage sensor domain of the silent Kv6.4 subunit" by Debanjan Tewari, Christian Sattler, and Klaus Benndorf

Thank you for submitting your revised Research Article to The Journal of Physiology. It has been assessed by the original Reviewing Editor and Referees and has been well received. Some final revisions have been requested.

REVISION CHECKLIST:

We look forward to receiving your revised submission.

Yours sincerely,

Peying Fong
Senior Editor
The Journal of Physiology

REQUIRED ITEMS FOR REVISION

- You must start the Methods section with a paragraph headed Ethical approval (https://jp.msubmit.net/cgi-bin/main.plex?form_type=display_requirements#methods).

Research must comply with The Journal's policies regarding animal experiments (<https://physoc.onlinelibrary.wiley.com/hub/animal-experiments>) and adherence to these policies must be stated in the manuscript.

Authors should confirm in their Methods section that their experiments were carried out according to the guidelines laid down by their institution's animal welfare committee, including an ethics approval reference number. The Methods section must contain a statement about access to food, water and housing, details of the anaesthetic regime: anaesthetic used, dose and route of administration, and method of killing the experimental animals.

- Papers must comply with the Statistics Policy: https://jp.msubmit.net/cgi-bin/main.plex?form_type=display_requirements#statistics.

In summary:

- If $n \leq 30$, all data points must be plotted in the figure in a way that reveals their range and distribution. A bar graph with data points overlaid, a box and whisker plot or a violin plot (preferably with data points included) are acceptable formats.
- If $n > 30$, then the entire raw dataset must be made available either as supporting information, or hosted on a not-for-profit repository, e.g. FigShare, with access details provided in the manuscript.
- 'n' clearly defined (e.g. x cells from y slices in z animals) in the Methods. Authors should be mindful of pseudoreplication.
- All relevant 'n' values must be clearly stated in the main text, figures and tables.
- The most appropriate summary statistic (e.g. mean or median and standard deviation) must be used. Standard Error of the Mean (SEM) alone is not permitted.
- Exact p values must be stated. Authors must not use 'greater than' or 'less than'. Exact p values must be stated to three significant figures even when 'no statistical significance' is claimed.

EDITOR COMMENTS

Reviewing Editor:

Methods Details:

The experiments use *Xenopus* for oocyte isolation. Additional details should be included in accordance with the J Physiol Guidelines. This includes euthanasia, anesthesia protocol/monitoring, post-operative/analgesia care, and a description of the isolation.

Comments to the Author:

The referees re-evaluated the manuscript. They agree that the authors have improved but also identify concerns limiting the manuscript's impact on the field. These concerns include but are not limited to, data analysis and presentation. Additionally, some details regarding the Oocyte isolation are needed. This includes additional methodological detail on euthanasia, feeding information, anesthesia protocol/monitoring, post-operative/analgesia care, and a brief description of the oocyte isolation protocol.

Senior Editor:

Review of your revised manuscript "Functional control of heteromeric Kv2.1/6.4 channels by the voltage sensor domain of the silent Kv6.4 subunit" is now complete. From the attached comments, you will read that both original Referees appreciate an obvious improvement in the manuscript and the effort taken to respond to queries raised in review of the initial submission. However, as summarized by the Reviewing Editor, there remains room for improvement in analysis and presentation. Specific points are detailed in each Referee's comments. Please do address each Referee's points fully in your revised manuscript.

Regarding Referee 2's query pertaining to The Journal of Physiology's standard for expression of p values, this can be found in the Information for Authors. I commend you to the following published statement of the Statistical Policy (https://jp.msubmit.net/cgi-bin/main.plex?form_type=display_requirements#statistics):

"P value for statistical tests

For a given conclusion to be assessed, the exact p values must be stated to three significant figures (not decimal places) even when 'no statistical significance' is being reported (i.e. for anything >0.001 , please report to 3 significant figures, e.g. 0.00236 or 0.523, etc.). These should be stated in the main text, figures and their legends and tables. The only exception to this is if p is less than 0.001, in which case '<' is permitted. Trend statements are not permitted (i.e. 'x increased, but was not significant'). Where there are many comparisons, a table of p values is requested. Asterisks alone should not be used to denote significance within figures."

While scientific notation is not explicitly prohibited, please note that exceptions are permitted in instances when $p < 0.001$. Therefore, in those instances (for example, in figure 2H and 2I, 4H and 4I, 7D and 7E, etc), please designate accordingly.

As requested by the Reviewing Editor, please do also include the details of vertebrate (*Xenopus laevis*) care, anesthesia, and euthanasia, as well as post-operative care.

Thank you for this contribution to The Journal of Physiology. We look forward to receiving your revised manuscript soon.

REFeree COMMENTS

Referee #1:

This insightful study has been nicely improved by revision, with many suggestions implemented thoughtfully. I do provide a few more suggestions and concerns related to author responses, that are each minor yet two of which seem important to address:

Reviewer #1 Suggestion 2 (important to address)

"... the dimer exhibited improved expression compared to expression of the monomers."

The statement seems incomplete, the results presented seem to show that the Kv 2.1_6.4, dimer exhibited improved expression compared to expression of the Kv 6.4_2.1 dimer or Kv 2.1 + 6.4 monomers.

Also, the data kindly provided to reviewers is not shown in the manuscript. Showing the data in the manuscript along with amounts of RNA injected would enhance credibility with readers.

Reviewer #1 Suggestion 3

The response nicely clarifies the proposal in the manuscript that "Kv6.4 subunits can partly activate the Kv2.1_6.4 channels on their own if the Kv2.1 subunits are blocked."

Yet it seems unlikely to me that Kv2.1_6.4 channels conduct while Kv2.1 subunits held in a resting conformation by GTX. My skepticism originates from the finding that Kv6.4 voltage sensors normally move at voltages far more negative than Kv2.1, yet no K⁺ conductance correlates with the voltage range of 6.4 voltage sensor movement (Bocksteins 2017 doi: 10.1038/srep41646) suggesting that Kv6.4 activation alone is insufficient to open the channel at all.

One way to possibly gain further insight into this might be to compare the kinetics of activation/deactivation of Kv2.1_6.4 + 500 nM GTX below 50 mV to Kv2.1_6.4 (no GTX). If they are different then the modified gating would support the proposal that Kv6.4 subunits can partly activate the Kv2.1_6.4 channels on their own if the Kv2.1 subunits are blocked.

Absent a good reason why it is not possible, I continue to think the manuscript would be improved by also mentioning the possibility that the residual current are from apo Kv2.1_6.4 channels.

Why not mention alternate possibilities?

Reviewer #1 Suggestion 4. (important to address)

Prefacing with a grave apology if I am somehow misguided, I cannot reconcile the lineshapes in figure

8C with actual single-exponential fits. I have tried scaling single-exponential fits to match and it does not work. I think there is a problem here.

Referee #2:

Major comments

The authors refer to individual panels (e.g. A, B, C) within each Figure only some of the time in the main text. For clarity and consistency, the authors need to edit the text, especially the Results section, to make sure each individual panel of each figure is referred to in the text.

For Figure 5 with GTX inhibition of control Kv2.1, are these dimers of concatenated dimers? If so, please edit the schematic of the tetramer to have linkers to indicate dimers of dimers. If not, how do the authors know that the additional component of GTX inhibition seen in 2.1_6.4 dimers is not a general property of linkers or dimers of dimers? In other words, does the additional component appear in Kv2.1-2.1 dimers?

Please describe how 'degree of CSI' is calculated. Is it $\text{final current} - \text{initial current} * 100$ during recovery?

Since the tau and degree of CSI are so different (in an interesting way!) I would suggest adding the exemplar current for 2.1-6.4L and 2.1-6.4W for comparison in Figure 7.

For consistency, I would suggest adding exemplar currents for the 2.1-6.4L and 2.1-6.4W dimer of dimers to Figure 8.

All the p values listed for scatter plots, e.g. Fig. 1K,L, are given with scientific notation e.g. 8.97×10^{-13} . Is this standard for J Physiol?

Minor comments

Line 312 of the text, is 'respective' the correct word?

In Figure 9, is there a way to draw the state diagram so that the arrows from I0 to C do not cross on top of the C* to Ic arrows?

END OF COMMENTS

Dear Dr Tewari,

Re: JP-RP-2025-288376R1 "Functional control of heteromeric Kv2.1/6.4 channels by the voltage sensor domain of the silent Kv6.4 subunit" by Debanjan Tewari, Christian Sattler, and Klaus Benndorf

Thank you for submitting your revised Research Article to The Journal of Physiology. It has been assessed by the original Reviewing Editor and Referees and has been well received. Some final revisions have been requested.

Your revised manuscript should be submitted online using the link in your Author Tasks <https://jp.msubmit.net/cgi-bin/main.plex?el=A3JS7GzS6B1hQQ2F4A9fdhrhVOZBnJYjJ0sw2nx6UiQZ>. This link is accessible via your account as Corresponding Author; it is not available to your co-authors. If this presents a problem, please contact journal staff (jp@physoc.org). Image files from the previous version are retained on the system. Please ensure you replace or remove any files that are being revised.

This will enable Authors to create and download high-resolution figures. If authors have

used the free BioRender service, they can use the instructions provided in the link above to download a high-resolution version suitable for publication.

LANGUAGE EDITING AND SUPPORT FOR PUBLICATION: If you would like help with English language editing, or other article preparation support, Wiley Editing Services offers expert help, including English Language Editing, as well as translation, manuscript formatting, and figure formatting at www.wileyauthors.com/eo/preparation. You can also find resources for Preparing Your Article for general guidance about writing and preparing your manuscript at www.wileyauthors.com/eo/prepresources.

REVISION CHECKLIST:

We look forward to receiving your revised submission.

Yours sincerely,

Peying Fong
Senior Editor
The Journal of Physiology

REQUIRED ITEMS FOR REVISION

- You must start the Methods section with a paragraph headed Ethical approval (https://jp.msubmit.net/cgi-bin/main.plex?form_type=display_requirements#methods).

Research must comply with The Journal's policies regarding animal experiments (<https://physoc.onlinelibrary.wiley.com/hub/animal-experiments>) and adherence to these policies must be stated in the manuscript.

Authors should confirm in their Methods section that their experiments were carried out according to the guidelines laid down by their institution's animal welfare committee, including an ethics approval reference number. The Methods section must contain a statement about access to food, water and housing, details of the anaesthetic regime: anaesthetic used, dose and route of administration, and method of killing the experimental animals.

A separate section has been added in the Materials and methods section entitled "Ethical Approval".

- Papers must comply with the Statistics Policy: https://jp.msubmit.net/cgi-bin/main.plex?form_type=display_requirements#statistics.

In summary:

- If n {less than or equal to} 30, all data points must be plotted in the figure in a way that reveals their range and distribution. A bar graph with data points overlaid, a box and whisker plot or a violin plot (preferably with data points included) are acceptable formats.

- If $n > 30$, then the entire raw dataset must be made available either as supporting information, or hosted on a not-for-profit repository, e.g. FigShare, with access details provided in the manuscript.
- 'n' clearly defined (e.g. x cells from y slices in z animals) in the Methods. Authors should be mindful of pseudoreplication.
- All relevant 'n' values must be clearly stated in the main text, figures and tables.
- The most appropriate summary statistic (e.g. mean or median and standard deviation) must be used. Standard Error of the Mean (SEM) alone is not permitted.
- Exact p values must be stated. Authors must not use 'greater than' or 'less than'. Exact p values must be stated to three significant figures even when 'no statistical significance' is claimed.

We have revised the manuscript to align with the Journal of Physiology's standards.

We sincerely appreciate the reviewers' thoughtful feedback and engagement with our work. In our revised manuscript, we have carefully addressed all the suggestions and critiques from both referees, as well as the comments from the Reviewing Editor and Senior Editor. All revisions are highlighted in yellow in the manuscript. Below, we provide detailed responses to each referee's comments along with the comments from the senior and reviewing editor in blue.

EDITOR COMMENTS

Reviewing Editor:

Methods Details:

The experiments use Xenopus for oocyte isolation. Additional details should be included in accordance with the J Physiol Guidelines. This includes euthanasia, anesthesia protocol/monitoring, post-operative/analgesia care, and a description of the isolation.

Comments to the Author:

The referees re-evaluated the manuscript. They agree that the authors have improved but

also identify concerns limiting the manuscript's impact on the field. These concerns include but are not limited to, data analysis and presentation. Additionally, some details regarding the Oocyte isolation are needed. This includes additional methodological detail on euthanasia, feeding information, anesthesia protocol/monitoring, post-operative/analgesia care, and a brief description of the oocyte isolation protocol.

Thank you for the re-evaluation and for providing valuable feedback. We appreciate the reviewers' acknowledgment of the improvements made and their thoughtful comments aimed at strengthening the manuscript's impact. We acknowledge the need for further methodological details regarding oocyte isolation. We have provided comprehensive information on euthanasia, feeding protocols, anesthesia administration and monitoring, post-operative care, analgesia, and a detailed description of the oocyte isolation procedure under the heading "Ethical Approval" in the Materials and methods section.

Senior Editor:

Review of your revised manuscript "Functional control of heteromeric Kv2.1/6.4 channels by the voltage sensor domain of the silent Kv6.4 subunit" is now complete. From the attached comments, you will read that both original Referees appreciate an obvious improvement in the manuscript and the effort taken to respond to queries raised in review of the initial submission. However, as summarized by the Reviewing Editor, there remains room for improvement in analysis and presentation. Specific points are detailed in each Referee's comments. Please do address each Referee's points fully in your revised manuscript.

Regarding Referee 2's query pertaining to The Journal of Physiology's standard for expression of p values, this can be found in the Information for Authors. I commend you to the following published statement of the Statistical Policy (https://jp.msubmit.net/cgi-bin/main.plex?form_type=display_requirements#statistics):

"P value for statistical tests

For a given conclusion to be assessed, the exact p values must be stated to three significant figures (not decimal places) even when 'no statistical significance' is being reported (i.e. for anything >0.001 , please report to 3 significant figures, e.g. 0.00236 or 0.523, etc.). These should be stated in the main text, figures and their legends and tables. The only exception to this is if p is less than 0.001, in which case ' $<$ ' is permitted. Trend statements are not

permitted (i.e. 'x increased, but was not significant'). Where there are many comparisons, a table of p values is requested. Asterisks alone should not be used to denote significance within figures."

While scientific notation is not explicitly prohibited, please note that exceptions are permitted in instances when $p < 0.001$. Therefore, in those instances (for example, in figure 2H and 2I, 4H and 4I, 7D and 7E, etc), please designate accordingly.

Thank you for pointing out the scientific notation. We have revised it to align with the Journal of Physiology's standards for the above-mentioned figures, as well as Figure 1, as requested by the second reviewer.

As requested by the Reviewing Editor, please do also include the details of vertebrate (*Xenopus laevis*) care, anesthesia, and euthanasia, as well as post-operative care.

As mentioned earlier we have included all the details in "Ethical Approval" in the Materials and methods section.

Thank you for this contribution to The Journal of Physiology. We look forward to receiving your revised manuscript soon.

REFEREE COMMENTS

Referee #1:

This insightful study has been nicely improved by revision, with many suggestions implemented thoughtfully. I do provide a few more suggestions and concerns related to author responses, that are each minor yet two of which seem important to address:

Reviewer #1 Suggestion 2 (important to address)

"... the dimer exhibited improved expression compared to expression of the monomers."

The statement seems incomplete, the results presented seem to show that the Kv 2.1_6.4, dimer exhibited improved expression compared to expression of the Kv 6.4_2.1 dimer or Kv 2.1 + 6.4 monomers. Also, the data kindly provided to reviewers is not shown in the manuscript. Showing the data in the manuscript along with amounts of RNA injected would enhance credibility with readers.

We sincerely thank the reviewer for highlighting the incompleteness in the statement. We have revised the statement accordingly, as suggested, and the changes are highlighted in the revised text. Additionally, we have included the expression profile of the three constructs—Kv2.1_6.4, Kv6.4_2.1, and Kv2.1/6.4—as a separate figure (Figure 4). As asked the mRNA concentration injected and oocyte incubation time are provided in the figure legend.

Reviewer #1 Suggestion 3

The response nicely clarifies the proposal in the manuscript that "Kv6.4 subunits can partly activate the Kv2.1_6.4 channels on their own if the Kv2.1 subunits are blocked."

Yet it seems unlikely to me that Kv2.1_6.4 channels conduct while Kv2.1 subunits held in a resting conformation by GTX. My skepticism originates from the finding that Kv6.4 voltage sensors normally move at voltages far more negative than Kv2.1, yet no K⁺ conductance correlates with the voltage range of 6.4 voltage sensor movement (Bocksteins 2017 doi: 10.1038/srep41646) suggesting that Kv6.4 activation alone is insufficient to open the channel at all. One way to possibly gain further insight into this might be to compare the kinetics of activation/deactivation of Kv2.1_6.4 + 500 nM GTX below 50 mV to Kv2.1_6.4 (no GTX). If they are different then the modified gating would support the proposal that Kv6.4 subunits can partly activate the Kv2.1_6.4 channels on their own if the Kv2.1 subunits are blocked. Absent a good reason why it is not possible, I continue to think the manuscript would be improved by also mentioning the possibility that the residual current are from apo Kv2.1_6.4 channels.

Why not mention alternate possibilities?

We agree with the reviewer's comment on the discrepancy of the Kv6.4 voltage sensor movement and activation of the whole channel in Kv2.1/6.4 wt channels (Bocksteins 2017). However, the reported shift of ~ 10 mV in the activation to hyperpolarized potentials and the shallower slope of the G-V relationship when the 6.4 subunit is incorporated compared to homomeric Kv2.1 channels, indicates that the energy for activation is lowered by the 6.4

subunit. The disabling effect of either the Leucin mutation in Kv2.1 subunit and the blocking with GTX-1E allows for further separate the contribution of the 6.4 and the 2.1 subunit in the activation process of the whole channel. The very strong effect of the toxin and the small remaining activation between 0 and +50 mV indicates a range where activation by only two 6.4 subunits is sufficient to open the channel. This partial activation is probably hidden in wildtype Kv2.1/6.4 channels because of the full function of Kv2.1 subunits.

We adapted the part in the manuscript to:

"The blocking effect of GTX-1E is even stronger than the disabling effect of the Leucin mutation in Kv2.1 subunits. This allows us to further dissect the function of the different subunits in heteromeric Kv2.1/6.4 channels. The absence of any activation in Kv2.1 channels in the presence of 500 nM GTX-1E until voltages of +50 mV and the small remaining activation between 0 and +50 mV indicates a range where partial activation by only two 6.4 subunits is sufficient to open the channel."

Reviewer #1 Suggestion 4. (important to address)

Prefacing with a grave apology if I am somehow misguided, I cannot reconcile the lineshapes in figure 8C with actual single-exponential fits. I have tried scaling single-exponential fits to match and it does not work. I think there is a problem here.

We appreciate the reviewer's concern regarding the single exponential fits and the line shape presented in Figure 8C (now 9E). To provide greater clarity, we have included the residuals of the fits as a separate panel in the figure with extended ordinate. The residuals confirm that the fit is accurate, demonstrating that the data aligns well with a single exponential.

Referee #2:

Major comments

The authors refer to individual panels (e.g. A, B, C) within each Figure only some of the time

in the main text. For clarity and consistency, the authors need to edit the text, especially the Results section, to make sure each individual panel of each figure is referred to in the text.

We appreciate the reviewer's suggestion regarding the reference to individual panels within each figure. To ensure clarity and consistency, we have carefully revised the text, particularly the Results section, to explicitly refer to each relevant panel in the figures.

For Figure 5 with GTX inhibition of control Kv2.1, are these dimers of concatenated dimers? If so, please edit the schematic of the tetramer to have linkers to indicate dimers of dimers. If not, how do the authors know that the additional component of GTX inhibition seen in 2.1_6.4 dimers is not a general property of linkers or dimers of dimers? In other words, does the additional component appear in Kv2.1-2.1 dimers?

We appreciate the reviewer's insightful question regarding Figure 5 (now figure 6) and the nature of the Kv2.1 constructs. The Kv2.1 channels in this experiment are not dimers of concatenated dimers. As asked by the reviewer we have included the data for Kv 2.1_2.1 dimers of concatenated dimers in figure 6.

Moreover, to clarify this, we have revised the schematic to more accurately represent the construct with linkers indicating dimers of dimers.

Regarding the additional component of GTX inhibition observed in Kv2.1_6.4 dimers, we have considered whether this effect could be a general property of linkers or dimeric constructs. To address this, we have analyzed Kv2.1_2.1 dimers and found that they do not exhibit the same additional component of GTX inhibition. This suggests that the observed effect is specific to Kv2.1_6.4 dimers rather than a general consequence of linker usage.

Please describe how 'degree of CSI' is calculated. Is it $\text{final current} - \text{initial current} * 100$ during recovery?

We thank the reviewer for highlighting this. The degree of CSI is calculated by subtracting the ratio of the initial current to the final current from one. We have now included a sentence which describes the calculation in the legend for Figure 7 (now Figure 8).

Since the tau and degree of CSI are so different (in an interesting way!) I would suggest adding the exemplar current for 2.1-6.4L and 2.1-6.4W for comparison in Figure 7.

We would like to thank the reviewer for mentioning the about the exemplar current traces. We have included the current traces in figure 7 (now figure 8)

For consistency, I would suggest adding exemplar currents for the 2.1-6.4L and 2.1-6.4W dimer of dimers to Figure 8.

The exemplar current traces for the Kv2.1-6.4L and 2.1_6.4W dimer of dimers have been included in figure 8 (now figure 9).

All the p values listed for scatter plots, e.g. Fig. 1K,L, are given with scientific notation e.g. 8.97×10^{-13} . Is this standard for J Physiol?

We thank the reviewer for highlighting the use of scientific notation in the figures. We have revised the notation to align with The Journal of Physiology standards.

Minor comments

Line 312 of the text, is 'respective' the correct word?

We have used "appropriate" in place of "respective".

In Figure 9, is there a way to draw the state diagram so that the arrows from I0 to C do not cross on top of the C* to Ic arrows?

We have revised the state diagram in accordance with the reviewer's comments.

Dear Dr Tewari,

Re: JP-RP-2025-288376R2 "Functional control of heteromeric Kv2.1/6.4 channels by the voltage sensor domain of the silent Kv6.4 subunit" by Debanjan Tewari, Christian Sattler, and Klaus Benndorf

Thank you for submitting your manuscript to The Journal of Physiology. It has been assessed by a Reviewing Editor and by 2 expert referees and we are pleased to tell you that it is acceptable for publication following satisfactory revision.

REVISION CHECKLIST:

We look forward to receiving your revised submission.

Yours sincerely,

Peying Fong
Senior Editor
The Journal of Physiology

REQUIRED ITEMS FOR REVISION

- You must start the Methods section with a paragraph headed Ethical approval (https://jp.msubmit.net/cgi-bin/main.plex?form_type=display_requirements#methods).

Research must comply with The Journal's policies regarding animal experiments (<https://physoc.onlinelibrary.wiley.com/hub/animal-experiments>) and adherence to these policies must be stated in the manuscript.

Authors should confirm in their Methods section that their experiments were carried out according to the guidelines laid down by their institution's animal welfare committee, including an ethics approval reference number. The Methods section must contain a statement about access to food, water and housing, details of the anaesthetic regime: anaesthetic used, dose and route of administration, and method of killing the experimental animals.

- Papers must comply with the Statistics Policy: https://jp.msubmit.net/cgi-bin/main.plex?form_type=display_requirements#statistics.

In summary:

- If $n \leq 30$, all data points must be plotted in the figure in a way that reveals their range and distribution. A bar graph with data points overlaid, a box and whisker plot or a violin plot (preferably with data points included) are acceptable formats.
- If $n > 30$, then the entire raw dataset must be made available either as supporting information, or hosted on a not-for-profit repository, e.g. FigShare, with access details provided in the manuscript.
- 'n' clearly defined (e.g. x cells from y slices in z animals) in the Methods. Authors should be mindful of pseudoreplication.
- All relevant 'n' values must be clearly stated in the main text, figures and tables.
- The most appropriate summary statistic (e.g. mean or median and standard deviation) must be used. Standard Error of the Mean (SEM) alone is not permitted.
- Exact p values must be stated. Authors must not use 'greater than' or 'less than'. Exact p values must be stated to three significant figures even when 'no statistical significance' is claimed.

EDITOR COMMENTS

Reviewing Editor:

The authors have satisfactorily addressed most concerns. However, referee one still has some minor but important edits described in their comments to the authors, and they require clarification on the exponential fit data reported for Figures 8E and 9E.

Senior Editor:

Your revised manuscript has been assessed by the original referees and the Reviewing Editor. As you will read, they concur that the changes incorporated into this version have improved the manuscript considerably. I commend to your attention two points raised by Referee 1 that must be corrected. Please can you address these? We look forward to receiving a corrected version of your manuscript.

REFeree COMMENTS

Referee #1:

This insightful study has been improved again by the second revision. However, there are still 2 types of statements in the manuscript that I think are incorrect:

Reviewer #1 Suggestion 3:

I continue to disagree with the unequivocal conclusions about GTX (which are featured in the abstract), as there are alternate explanations (discussed in prior reviews). Here simple softening of wording could satisfy this concern:

"Moreover, the specific Kv2.1 blocker guangxitoxin unmasks that Kv6.4 subunits can partly activate Kv2.1_6.4 channels."

change to:

"Moreover, results with the specific Kv2.1 blocker guangxitoxin suggest that Kv6.4 subunits may partly activate Kv2.1_6.4 channels."

"The absence of any activation in Kv2.1 channels in the presence of 500 nM GTX-1E until voltages of 50 mV and the small remaining activation between 0 and +50 mV indicates a range where partial activation by only two 6.4 subunits is sufficient to open the channel."

change to:

"The absence of any activation in Kv2.1 channels in the presence of 500 nM GTX-1E until voltages of 50 mV and the small remaining activation between 0 and +50 mV could indicate a range where partial activation by only two 6.4 subunits is sufficient to open the channel."

Reviewer #1 Suggestion 4:

I think there is an error. I still cannot reconcile the lineshapes in figure 8E and 9E with simple single monoexponential fits. While the residuals look fine, it is the fit functions that have produced the lines themselves that raise concern. An appropriate single exponential fit function is:

$$y(t)=A e^{(-t/\tau)} + C$$

I think some other function is plotted in figure 8E and 9E.

To reconcile, the manuscript could state the actual fit function used or fit with the function I've provided. Again, if I'm somehow missing something, I apologize for the trouble.

Referee #2:

The authors have adequately responded to my previous concerns.

END OF COMMENTS

Dear Dr Tewari,

Re: JP-RP-2025-288376R2 "Functional control of heteromeric Kv2.1/6.4 channels by the voltage sensor domain of the silent Kv6.4 subunit" by Debanjan Tewari, Christian Sattler, and Klaus Benndorf

Thank you for submitting your manuscript to The Journal of Physiology. It has been assessed by a Reviewing Editor and by 2 expert referees and we are pleased to tell you that it is acceptable for publication following satisfactory revision.

LANGUAGE EDITING AND SUPPORT FOR PUBLICATION: If you would like help with English

language editing, or other article preparation support, Wiley Editing Services offers expert help, including English Language Editing, as well as translation, manuscript formatting, and figure formatting at www.wileyauthors.com/eeo/preparation. You can also find resources for Preparing Your Article for general guidance about writing and preparing your manuscript at www.wileyauthors.com/eeo/prepresources.

REVISION CHECKLIST:

We look forward to receiving your revised submission.

Yours sincerely,

Peying Fong
Senior Editor
The Journal of Physiology

REQUIRED ITEMS FOR REVISION

- You must start the Methods section with a paragraph headed Ethical approval (https://jp.msubmit.net/cgi-bin/main.plex?form_type=display_requirements#methods).

Research must comply with The Journal's policies regarding animal experiments (<https://physoc.onlinelibrary.wiley.com/hub/animal-experiments>) and adherence to these policies must be stated in the manuscript.

Authors should confirm in their Methods section that their experiments were carried out according to the guidelines laid down by their institution's animal welfare committee, including an ethics approval reference number. The Methods section must contain a statement about access to food, water and housing, details of the anaesthetic regime: anaesthetic used, dose and route of administration, and method of killing the experimental animals.

- Papers must comply with the Statistics Policy: https://jp.msubmit.net/cgi-bin/main.plex?form_type=display_requirements#statistics.

In summary:

- If $n \leq 30$, all data points must be plotted in the figure in a way that reveals their range and distribution. A bar graph with data points overlaid, a box and whisker plot or a violin plot (preferably with data points included) are acceptable formats.

- If $n > 30$, then the entire raw dataset must be made available either as supporting information, or hosted on a not-for-profit repository, e.g. FigShare, with access details provided in the manuscript.

- 'n' clearly defined (e.g. x cells from y slices in z animals) in the Methods. Authors should be mindful of pseudoreplication.

- All relevant 'n' values must be clearly stated in the main text, figures and tables.

- The most appropriate summary statistic (e.g. mean or median and standard deviation) must be used. Standard Error of the Mean (SEM) alone is not permitted.

- Exact p values must be stated. Authors must not use 'greater than' or 'less than'. Exact p values must be stated to three significant figures even when 'no statistical significance' is claimed.

We sincerely appreciate the reviewers' thoughtful feedback and engagement with our work. In our revised manuscript, we have carefully addressed all the suggestions and critiques from both referees, as well as the comments from the Reviewing Editor and Senior Editor. All revisions are highlighted in yellow in the manuscript. Below, we provide detailed responses to each referee's comments along with the comments from the senior and reviewing editor in blue.

EDITOR COMMENTS

Reviewing Editor:

The authors have satisfactorily addressed most concerns. However, referee one still has some minor but important edits described in their comments to the authors, and they require clarification on the exponential fit data reported for Figures 8E and 9E.

Thank you for your thoughtful feedback and for sharing the reviewers' latest comments.

We are encouraged that all concerns have been resolved and greatly appreciate Referee 1's careful attention to the remaining issues. We have included the minor edits as requested and provided the necessary clarification regarding the exponential fits for Figures 8E and 9E.

Senior Editor:

Your revised manuscript has been assessed by the original referees and the Reviewing Editor. As you will read, they concur that the changes incorporated into this version have improved the manuscript considerably. I commend to your attention two points raised by Referee 1 that must be corrected. Please can you address these? We look forward to receiving a corrected version of your manuscript.

Thank you for your positive feedback and for coordinating the review process. We are pleased to hear that the revisions have improved the manuscript considerably. We appreciate Referee 1's additional comments. We have addressed them and submitted an improved version of the manuscript.

REFEREE COMMENTS

Referee #1:

This insightful study has been improved again by the second revision. However, there are still 2 types of statements in the manuscript that I think are incorrect:

Reviewer #1 Suggestion 3:

I continue to disagree with the unequivocal conclusions about GTX (which are featured in the abstract), as there are alternate explanations (discussed in prior reviews). Here simple softening of wording could satisfy this concern:

"Moreover, the specific Kv2.1 blocker guangxitoxin unmasks that Kv6.4 subunits can partly activate Kv2.1_6.4 channels."

change to:

"Moreover, results with the specific Kv2.1 blocker guangxitoxin suggest that Kv6.4 subunits may partly activate Kv2.1_6.4 channels."

"The absence of any activation in Kv2.1 channels in the presence of 500 nM GTX-1E until voltages of 50 mV and the small remaining activation between 0 and +50 mV indicates a range where partial activation by only two 6.4 subunits is sufficient to open the channel."

change to:

"The absence of any activation in Kv2.1 channels in the presence of 500 nM GTX-1E until voltages of 50 mV and the small remaining activation between 0 and +50 mV could indicate a range where partial activation by only two 6.4 subunits is sufficient to open the channel."

We sincerely thank the reviewer and softened the statement. The changes are highlighted in the abstract as well as in the text.

Reviewer #1 Suggestion 4:

I think there is an error. I still cannot reconcile the line shapes in figure 8E and 9E with simple single mono exponential fits. While the residuals look fine, it is the fit functions that have produced the lines themselves that raise concern. An appropriate single exponential fit function is:

$$y(t)=A e^{(-t/\tau)} + C$$

I think some other function is plotted in figure 8E and 9E.

To reconcile, the manuscript could state the actual fit function used or fit with the function I've provided. Again, if I'm somehow missing something, I apologize for the trouble.

We re-checked the fits of all recovery time courses and noticed that our routine indeed did a biexponential instead of a mono exponential fit. So, we repeated all fits with only one exponential and also replaced Figure 8E and 9E. One can identify this by slightly deviating curves. None of the conclusions in our manuscript is affected. We do like to apologize for this bad mistake and express our thanks to the reviewer for insisting with his criticism.

The manuscript reads now:

The recovery from inactivation was determined by fitting the exponential function $y = y_0 + A_1 \cdot (1 - \exp(-t/\tau_{rec}))$ to the recovery time courses using OriginPro 2019, where y is the normalized amplitude of the test current, y_0 a time-independent component, t the time and τ_{rec} the recovery time constant.

Referee #2:

The authors have adequately responded to my previous concerns.

Thank you for the re-evaluation and for providing valuable feedback. We appreciate the reviewers' acknowledgment of the improvements made and their thoughtful comments aimed at strengthening the manuscript's impact.

Dear Dr Tewari,

Re: JP-RP-2025-288376R3 "Functional control of heteromeric Kv2.1/6.4 channels by the voltage sensor domain of the silent Kv6.4 subunit" by Debanjan Tewari, Christian Sattler, and Klaus Benndorf

We are pleased to tell you that your paper has been accepted for publication in The Journal of Physiology.

Yours sincerely,

Peying Fong
Senior Editor
The Journal of Physiology

If you would like to receive our 'Research Roundup', a monthly newsletter highlighting the cutting-edge research published in The Physiological Society's family of journals (The Journal of Physiology, Experimental Physiology, Physiological Reports, The Journal of Nutritional Physiology and The Journal of Precision Medicine: Health and Disease), please click this link, fill in your name and email address and select 'Research Roundup':
<https://www.physoc.org/journals-and-media/membernews>

- You can help your research get the attention it deserves! Check out Wiley's free Promotion Guide for best-practice recommendations for promoting your work at: www.wileyauthors.com/eoo/guide. You can learn more about Wiley Editing Services which offers professional video, design, and writing services to create shareable video abstracts, infographics, conference posters, lay summaries, and research news stories for your research at: www.wileyauthors.com/eoo/promotion.

EDITOR COMMENTS

Reviewing Editor:

The authors have addressed each one of Referee 1's concerns and modified the manuscript accordingly.

Senior Editor:

Thank you for responding to the remaining points raised during review of your revised manuscript. The Reviewing Editor and I concur that all changes incorporated are appropriate and adequately address all residual concerns, in particular Referee 1's fourth point regarding the data fit.

At this time, I am pleased to congratulate you on this interesting study and thank you for submitting your work to The Journal of Physiology. I look forward to seeing it in its final, published form.